# CHESS-SCAPE: High resolution future projections of multiple climate scenarios for the United Kingdom derived from downscaled UKCP18 regional climate model output

Emma L. Robinson[1], Chris Huntingford[1], Valyaveetil Shamsudheen Semeena[1], and James M. Bullock[1]

[1]UK Centre for Ecology & Hydrology, Wallingford, UK

**Correspondence:** Emma L. Robinson (emrobi@ceh.ac.uk)

**Abstract.** In order to effectively model the potential impacts of future climate change, there is a requirement for climate data inputs which are a) high spatial and temporal resolution; b) explore a range of future climate change scenarios; c) are consistent with historical observations in the historical period; and d) provide an exploration of climate model uncertainty. This paper presents a suite of climate projections for the United Kingdom that conform to these requirements: CHESS-SCAPE.

CHESS-SCAPE is a 1 km resolution dataset containing eleven near-surface meteorological variables that can be used to as input to many different impacts models. The variables are available at several time resolutions, from daily to decadal means, for the years 1980 – 2080. It was derived from the state-of-the art regional climate projections in the United Kingdom Climate Projections 2018 (UKCP18) Regional Climate Model (RCM) UKCP18 12 km ensemble, downscaled to 1 km using a combination of physical and empirical methods to account for local topographic effects. CHESS-SCAPE has four ensemble
members, which were chosen to span the range of temperature and precipitation change in the UKCP18 ensemble, representing the ensemble climate model uncertainty.

CHESS-SCAPE consists of projections for four emissions scenarios, given by the Representative Concentration Pathways: RCP2.6, 4.5, 6.0 and 8.5, which were derived from the UKCP18 RCM RCP8.5 scenarios using time shifting and pattern scaling. These correspond to UK annual warming projections of between 0.9 – 1.9 K for RCP2.6 up to 2.8 – 4.3 K for RCP8.5 between
1980 – 2000 and 2060 – 2080. Little change in annual precipitation is projected, but larger changes in seasonal precipitation are seen with some scenarios projecting large increases in precipitation in the winter (up to 22 %) and large decreases in the summer (up to -39%). All four RCP scenarios and ensemble members are also provided with bias-correction, using the CHESS-met historical gridded dataset as a baseline.

With high spatial and temporal resolution, extensive range of warming scenarios and multiple ensemble members, CHESS-
SCAPE provides a comprehensive data resource for modellers of climate change impacts in the UK. The CHESS-SCAPE data are available for download from the NERC EDS Centre for Environmental Data Analysis: http://dx.doi.org/10.5285/8194b416cbee482b89e0dfbe17c5786c (Robinson et al., 2022).

# 1 Introduction

Climate projections obtained from climate model output are widely used to explore possible future scenarios, study climate change impacts, and make policy decisions in a wide variety of fields (IPCC, 2022). In general, due to the computational resources required, global climate models (GCMs) and earth system models (ESMs) are run at relatively coarse resolutions (tens to hundreds of km). However, in order to investigate and project climate change impacts regionally, higher spatial and temporal resolutions of climate data are necessary (Barros et al., 2014). Modelling of future floods, droughts and water resources requires

high enough spatial and temporal resolution to resolve local hydrological processes that are unresolvable in coarser resolution simulations (Wood et al., 2011; Bierkens et al., 2015; Melsen et al., 2016). Higher resolutions enable better partitioning of energy and water in land surface models, which is in turn essential for modelling carbon budgets and other biogeochemical processes (Ciais et al., 2014). Resolutions close to field scale are important for agricultural modelling (Karthikeyan et al., 2020) and food security (Betts et al., 2018). Analysis of biodiversity responses requires climate projections at a fine spatial and tem-

poral resolution in order to represent biological processes and species dynamics accurately, and to plan adaptation strategies (Nadeau et al., 2017; Dupont-Doaré and Alagador, 2021).

Since the start of numerical weather forecasting, and throughout the past seven decades, there has been an ongoing increase in the spatial resolution of the calculations. However, while this allows present-day forecasts to integrate equations on a resolution of order kilometres, this is still not possible for climate projections, due to their need to estimate evolving atmospheric

conditions over periods of centuries rather than days or hours. For this reason, most GCMs and ESMs have a spatial resolution of many tens or even hundreds of kilometres. Some higher resolution climate calculations exist, either from running GCMs or ESMs over shorter "timeslices" or nesting Regional Climate Models (RCMs) inside them. The recent sixth IPCC report (IPCC, 2021) points to multiple examples where increased resolution improves the representation of weather features. Bock et al. (2020) find that the newer generation of higher resolution ESMs removes many long-standing temperature and rainfall

projections biases. Specific improvements include, for instance, Priestley et al. (2020) who find a dependence of simulated storm tracks on resolution, and when this work is extended illustrates that higher resolution models better capture the structure of cyclones away from the tropics (Priestley and Catto, 2022). Confirmation exists that increased spatial resolution improves the statistical properties of rainfall patterns, such as for the tropics (Roberts et al., 2020). An overview of the benefits of higher resolution in simulated climate is presented in (Roberts et al., 2018). There are many regions globally where convective rain-

fall is more prevalent than large scale frontal rainfall, but to explicitly resolve the features of convective storms (rather than attempting parameterisation of their mean properties) needs a further step-change in resolution to kilometre scale. Prein et al. (2015) make a case for the benefits of convection permitting models (CPMs), which are now available for limited timeframes and geographical extents, but their intense computational requirements currently preclude transient projections corresponding to evolving rising atmospheric GHG concentrations over periods of decades (Slingo et al., 2022).

When kilometer scale climate projections are required, but available computational resources preclude running a CPM, an alternative is to run a coarser resolution model (GCM, ESM or RCM) and statistically downscale the climate data to the required high resolutions (eg Navarro-Racines et al., 2020; Rudd and Kay, 2016). Downscaling methods of varying complexity

can be used. Simple spatial interpolation is of use to be able to run impacts models at these resolutions, but it does not add any new high-resolution information. More useful are more complex methods which improve the modelling of sub-grid-scale

processes (Barros et al., 2014).

No matter the spatial resolution, most climate models run at a high temporal resolution (of order minutes to hours), however storage limitations often preclude storing outputs at higher than a daily timestep. In recent years, modelling efforts have begun to provide sub-daily outputs, including CMIP6 3-hourly and 6-hourly outputs (Eyring et al., 2016) and hourly outputs of the convection-permitting runs of UKCP18 (Fosser et al., 2019). However, providing this requires access to significant computa-

tional infrastructure to store and disseminate these data. For CHESS-SCAPE, the immediate needs of impacts modellers were balanced against the available processing and storage capabilities, with daily outputs being deemed the most appropriate.

A further requirement for impacts modelling is the ability to explore different scenarios of emissions and climate change (Ranasinghe et al., 2021; COP21, 2015). A widely used set of scenarios is the Representative Concentration Pathways (RCPs; van Vuuren et al., 2011a), which were introduced by the community at the request of the Intergovernmental Panel on Climate

Change (IPCC) after the Fourth Assessment Report (AR4; IPCC, 2007a), in order to provide input to newer climate models and explore the impacts of different climate policies (IPCC, 2007b). The RCPs provide a range of possible future concentration and emission scenarios of greenhouse gases and other pollutants, based on differing future policy decisions and mitigations (Meinshausen et al., 2011). They were used to force GCM and ESM contributions to the fifth phase of the IPPC's Coupled Model Intercomparison Project (CMIP5; Taylor et al., 2012) which informed the Fifth Assessment Report (AR5; IPCC, 2014).

There are four scenarios, RCP2.6, RCP4.5, RCP6.0 and RCP8.5, where the number refers to the resulting radiative forcing by the end of the 21st century in W m$^{-2}$. RCP2.6 is a mitigation scenario, aiming to limit global mean temperature increase to less than 2 °C (van Vuuren et al., 2011b), while RCP8.5 is a high emissions scenario with no climate change mitigation target (Riahi et al., 2011). RCP4.5 and 6.0 are medium stabilisation scenarios (Thomson et al., 2011; Masui et al., 2011), with similar total emissions and radiative forcing for the first half of the 21st century, but RCP4.5 stabilising sooner.

As part of the same process, efforts were also made by the IPCC to develop socio-economic scenarios — Shared Socioeconomic Pathways (SSPs) — that are complementary to the physical concentration scenarios (O'Neill et al., 2014, 2017). These are based on five alternative narratives of socio-economic development and their implications on energy, land use and emissions (Riahi et al., 2017). The SSPs and RCPs were used together in a scenario matrix to force models in a consistent way for CMIP5 (van Vuuren et al., 2014). This was built on to create combined socio-economic and physical scenarios for the sixth

phase of the IPPC's Coupled Model Intercomparison Project (CMIP6; Eyring et al., 2016; O'Neill et al., 2016). In order to facilitate regional impacts modelling in the UK, Pedde et al. (2021) have enriched and downscaled the global SSPs by using stakeholder co-production to integrate national knowledge to create consistent UK-specific scenarios, the UK-SSPs. These can be used alongside the CHESS-SCAPE data to consistently model the combined impacts of climate and socioeconomic change (eg Brown et al., 2022).

The range of possible socio-economic and emissions scenarios is one source of uncertainty in climate projections, however, even with the same emissions scenarios, different climate models produce different projections of climate variables in the future (Hawkins and Sutton, 2009; Wilby and Dessai, 2010). This is due to two other sources of uncertainty: climate model

uncertainty, differences in the choices of parameter values and the representation or approximation of processes (Hawkins and Sutton, 2009; Wilby and Dessai, 2010); and internal climate variability, natural fluctuations in the climate system due to its chaotic nature (Jain et al., 2023). Model intercomparions such as CMIP, where models are run with identical forcings, allow direct comparisons of different climate models to provide an estimate of the model uncertainty, while many contributers also run several realisations of the same scenario to understand internal climate variability (Taylor et al., 2012; Eyring et al., 2016). It is important to understand the uncertainty in the climate projections as part of the overall uncertainty of climate change impacts modelling (Ashraf Vaghefi et al., 2019; Schwarzwald and Lenssen, 2022).

Finally, climate models do not necessarily replicate the historical period exactly. This may be due to internal climate variability, with natural fluctuations lead to different realisations of the climate under the same forcings. It may also be due to the model uncertainty in the representation of earth system processes in the historical period. The latter is considered a "bias" compared to observations and bias correction is often applied to remove or reduce these biases (Teutschbein and Seibert, 2012), however the internal variability is an inherent property of the climate system and ideally should not be removed (Ayar et al., 2021). However, in practice the two are hard to distinguish, particularly in a small ensemble, so in practice bias-correction removes the differences introduced by both.

A variety of bias correction methods of varying complexity exist, from simple methods that adjust the mean (eg. linear scaling (Widmann et al., 2003)), to more complex methods that also adjust the distribution of variables (eg. quantile mapping (Leander and Buishand, 2007)). These are carried out by comparing the climate model output run for a historical period with observations in that period, calculating a correction based on this, then applying that correction to the whole of the climate model output. There must therefore be a sufficient overlap between the climate model output and observations and the observations must be of sufficient quality. Bias correction involves the strong assumption that the same biases will be preserved in the future, which has led to some criticism of the use of bias correction (Maraun, 2016; Ehret et al., 2012). However, for many instances of impacts modelling, raw climate model output is not sufficient if it is biased.

For process-based and empirical modelling studies, there are thus several ideal requirements for the input climate data: transient realisations of climate for the whole period of interest; field/landscape scale resolution (a few km); temporal resolution from 1 day to decadal means; a range of emissions scenarios; consistency with observations in the historical period; an exploration of climate model uncertainty. Although it is possible, and sometimes necessary, to relax some or all of these requirements, this paper describes the production of a climate dataset that attempts to fulfil all of these for impacts modelling in the United Kingdom (UK): CHESS-SCAPE (Robinson et al., 2022). This dataset can be used in combination with the United Kingdom Socioeconomic Scenarios (UK-SSPs; Pedde et al., 2021).

The CHESS-SCAPE dataset was created by statistically downscaling a subset of four members of the United Kingdom Climate Projections 2018 (UKCP18) 12 km resolution RCM output to 1 km, using an updated version of the existing CHESS methodology (Robinson et al., 2017). As well as interpolating the data from coarser to finer resolution, the CHESS methodology employs physical and empirical corrections to represent topography and other spatially varying effects on meteorology. Thus this dataset makes use of the increased relatively high resolution and regional parameterisation of the UKCP18 12 km RCM ensemble, with additional downscaling that further improves the modelling of climate change impacts below this resolution.

This also makes it consistent with the resolution of many existing historical gridded UK climate datasets, including CEH-GEAR (Tanguy et al., 2019), CHESS-met (Robinson et al., 2020a, b) and HadUK-Grid (Met Office et al., 2021).

As well as downscaling the climate model output, this dataset also applies bias-correction using the CHESS-met historical gridded data as a reference dataset. This ensures consistency with the historical period. Both bias-corrected and downscaled-only variables are provided in the final dataset.

The UKCP18 12 km RCM ensemble was only run for the RCP8.5 scenario. However, it is also useful to explore future scenarios where climate change mitigation policies have been implemented, particularly in tandem with the range of SSPs. To this end we have derived three other scenarios corresponding to RCP2.6, 4.5 and 6.0 by making use of the climate change patterns in the RCP8.5 data and global scenarios given by CMIP5 climate models (Taylor et al., 2012).

CHESS-SCAPE contains eleven near-surface climate variables at high spatial (1 km) resolution and a variety of temporal resolutions (from daily to decadal means) for the years 1980–2080 over the UK. The dataset contains four ensemble members and four emissions scenarios, giving a total of sixteen realisations of future climate, both with and without bias-correction. CHESS-SCAPE is a comprehensive set of climate projections suitable for modelling a wide range of climate change impacts.

In Section 2 we describe the input datasets used to create CHESS-SCAPE. In Section 3 we give a brief description of the methodology, with a full description in Appendices A, B and C. In Section 4 we describe the CHESS-SCAPE dataset and give a comparison with UKCP18.

## 2  Input data

### 2.1  UKCP18 RCM 12 km data

UKCP18 is the state of the art of climate projections for the UK. The full dataset consists of several "strands" of climate projections, covering a range of temporal coverage and spatial resolutions and a variety of methodologies (Murphy et al., 2018). It includes an ensemble of GCM output, high-resolution RCM output and probabilistic projections of regional climate change. The climate model used in UKCP18 was run at several resolutions, using as a base a perturbed parameter ensemble (PPE) of the Hadley Centre model with configuration HadGEM3-GC3.05. This is an immediate precursor of the HadGEM3-GC3.1 configuration, which was used for the Met Office Hadley Centre's GCM contributions to CMIP6. UKCP18 ran this PPE at a global scale (hereafter referred to as the global climate model PPE, GCM-PPE), then at a regional scale (regional climate model PPE, RCM-PPE) nested in the global runs (Murphy et al., 2018), and finally at a regional scale with high enough resolution (2.2 km) to allow free convection (convection permitting model PPE, CPM-PPE) (Fosser et al., 2019). The global ensemble was extended by including several members of the CMIP5 ensemble (Murphy et al., 2018).

The CPM-PPE runs are the highest-resolution and capture physics and extremes that are not available at lower resolutions, but computational limitations meant they were not run continuously for the whole projection period of UKCP18. Instead they were run for several 20-year periods, with 20-year gaps between (Fosser et al., 2019). While this is useful for impacts modelling in those time periods, when modelling long term dynamics and trends, the full time period is required, particularly when combined with transient socioeconomic scenarios, such as the UK-SSPs (Pedde et al., 2021).

All of the strands of UKCP18 provide output for RCP8.5 scenarios, but due to computational limitations, only a subset of the strands were used to explore alternative scenarios. The probabilistic strand made use of all four RCPs, plus the earlier SRES A1B scenario (Nakicenovic and Swart, 2000), and provides these as probability distributions of change relative to the baseline (Murphy et al., 2018). However, they are not spatially coherent, as this strand applies extensive statistical processing
to calculates probability distributions independently for each grid box (Murphy et al., 2018). They also do not provide transient output that could be used to run process-based models. As part of UKCP18, they created "derived projections", which are 60 km resolution projections for RCP2.6 globally plus two 50-year scenarios of constant 2 °C and 4 °C warming above pre-industrial for the UK at the same resolution. UKCP18 produced these by applying time shifting and pattern scaling to the GCM-PPE RCP8.5 output, except for the ensemble members that were derived from CMIP5, which had RCP2.6 scenarios
available (Gohar et al., 2018). Although these therefore provide a range of scenarios, the resolution is coarse compared to that required for impacts modelling.

In order to balance the advantages of higher resolution regional modelling against the limited time coverage of the CPM-PPE runs, we chose to use the 12 km resolution RCM-PPE (Met Office Hadley Centre, 2019) as the basis of CHESS-SCAPE. UKCP18 ran each RCM-PPE ensemble member nested inside the corresponding GCM-PPE ensemble member, at a higher
resolution than the GCM-PPE and with regional parameterisation over Europe for the entire 1981 – 2080 study period (Murphy et al., 2018). The native resolution was 0.11 °(approximately 12 km) on a rotated pole grid. They then regridded this to 12 km resolution on the Ordnance Survey Great Britain (OSGB) grid (EPSG, a) over the UK for publication using an area-weighted method (Met Office, 2018). The UKCP18 RCM-PPE includes 14 variables at a daily, monthly, seasonal and annual timestep, as well as timeslice means. We use the daily mean variables on the 12 km OSGB grid as the input for the CHESS-SCAPE
dataset. The full list of variables used is in Table 1.

### 2.1.1 Orography

In UKCP18, the RCM-PPE variables were modelled at an elevation, $z^U$ (m) which they derived from digital terrain mapping (European Environment Agency, 2021), then smoothed appropriately for use as the lower boundary condition of the atmospheric model. After they ran the model they then regridded the elevation to the 12 km grid aligned with OSGB for distribution
with the climate variables (Met Office Hadley Centre). Note that this elevation was not taken into consideration when regridding the climate variables to the 12 km OSGB grid (Met Office, 2018).

### 2.2 Ancillary 1 km information

As is described in Section 3.2, we interpolated the input variables and adjusted them for local topographic effects. In order to do this, we required several ancillary datasets at the target 1 km resolution. Some ancillary variables were available separately
for Great Britain (GB) and Northern Ireland (NI). Where necessary we combined these both onto the same OSGB grid. Most variables simply required the grid box elevation, which was calculated from digital terrain models (Sections 2.2.1 and 2.2.2). To downscale precipitation we made use of a historical observation-based dataset to derive the relationship between the 1 km and 12 km resolution gridded precipitation (Section 2.3). Although this is derived from historical data, we assumed that the

relationships are preserved into the future. For the wind speed we used a modelled mean wind speed dataset to relate the 1 km wind speeds to the 12 km means (Section 2.4). To calculate downwelling shortwave (SW) radiation required an albedo dataset, which again was based on historical data and assumed to apply into the future (Section 2.5).

### 2.2.1 Integrated Hydrological Digital Terrain Model

For GB, we used the elevation from the 50 m resolution Integrated Hydrological Digital Terrain Model (IHDTM; Morris et al., 1990). We calculated the 1 km elevation by taking the mean of all 50 m points in each 1 km grid box. We also calculated the aspect and slope of each 50 m grid box following the method of Horn (1981). This uses a weighted mean of the central differences between the adjacent points to find the north-south and east-west slope for each grid box. We then computed the aspect and slope of each 1 km grid box by calculating the mean angle.

### 2.2.2 Ordnance Survey of Northern Ireland Digital Terrain Model

For NI, the elevation was taken from the 50 m resolution Ordnance Survey of Northern Ireland Digital Terrain Model (OSNI DTM; Ordnance Survey of Northern Ireland, 2019). This is produced on the Irish grid (EPSG, b), which is not coincident with the OSGB grid. We found the 1 km elevation on the OSGB grid by calculating the mean of the OSNI points that were contained in each target 1 km grid box. Again, we calculated the aspect and slope by applying the method of Horn (1981) to the 50m resolution data, then calculating the mean angle of the points in each OSGB 1 km grid box.

### 2.3 Standardised Annual Average Rainfall (1961-90)

To downscale the precipitation, we used the Standardised Annual Average Rainfall (1961-90) dataset (SAAR; Spackman, 1993). This is a dataset of standardised precipitation at 1 km resolution based on observations and empirically corrected for the elevation of each grid box. It is derived from the UK Met Office's station network, which defines rainfall to be "the amount of liquid precipitation plus the liquid equivalent of any solid precipitation" (Sunter, 2020), so it is calculated with and applied to total precipitation without different treatments for different precipitation types. It is provided for the whole UK, but it is split into separate files for GB and NI. Over GB, the SAAR is provided on a 1 km grid that is offset by 500 m from the target CHESS grid, and the NI SAAR is provided on the Irish grid, so we interpolated the values to the target 1 km grid. We also require the SAAR at 12 km resolution, so we calculated the 12 km mean of the 1 km resolution variable.

### 2.4 UK Energy Technology Support Unit mean wind speed

We use the UK Energy Technology Support Unit (ETSU) mean wind speed dataset (Burch and Ravenscroft, 1992; Newton and Burch, 1985) to downscale the wind speed. This is a dataset of modelled mean wind speeds, calculated using Numerical Objective Analysis Boundary Layer modelling with a 1 km resolution model of the topography of the UK.

## 2.5 GlobAlbedo

For radiation calculations, we used GlobAlbedo (European Space Agency, 2011), an albedo dataset derived from European satellite observations (Muller et al., 2012). This provides monthly estimates of Bi-Hemispherical Reflectances (BHR, or "white-sky" albedo) and Direct Hemispherical Reflectances (DHR, or "black-sky" albedo) for the years 1998–2011. It is at a resolution of 30 arcsecs (approximately 1 km), which we reprojected to the OSGB grid, and used to create a mean-monthly climatology of both BHR and DHR to be used for all years of the projections. Future changes in albedo due to land use change and projected reductions in snow cover are not included in this work.

## 3 Methodology

This Section gives an overview of the methodology, full details are available in the Appendices. After identifying the UKCP18 RCM-PPE ensemble members to use (Section 3.1, the first step was to downscale the data from 12 km to 1 km (Section 3.2 and Appendix A). Next bias correction was applied to each ensemble member individually (Section 3.3 and Appendix B). Finally, alternative warming scenarios were populated, corresponding to the RCPs, using time shifting and pattern scaling (Section 3.4 and Appendix C).

### 3.1 Choice of sub-ensemble

The UKCP18 RCM-PPE is a perturbed parameter ensemble containing 12 members. Ensemble member 01 (EM01) was produced using the default configuration of the model (a regional version of the HadGEM-GC3.05 configuration). The other ensemble members used perturbations of parameters relating to the atmosphere, the land surface and aerosol components (Sexton et al., 2021). In addition, different trajectories of atmospheric $CO_2$ concentration were associated with each ensemble member. EM01 used concentrations prescribed in RCP8.5 for concentration driven runs, while the other ensemble members used $CO_2$ concentrations that were calculated by selected CMIP5 emissions-driven ensemble members. The combination of parameter perturbations and differing $CO_2$ concentrations resulted in a range of warming scenarios and impacts on UK climate (Murphy et al., 2018).

Due to storage and computational constraints, we deemed it impractical to downscale the entire UKCP18 RCM-PPE, so we chose four ensemble members to form the CHESS-SCAPE ensemble. We retained EM01, as it is the default model configuration, and chose three others in order to span the range of futures in the RCM-PPE. We based the choice on the mean change in UK near-surface air temperature and precipitation between the baseline period 1980 – 2000 and the final twenty years of the future projections 2060 – 2080 (Table 4). We calculated the change in the UK mean annual and seasonal air temperature and precipitation between the two periods. These are shown in Fig. 1. The smallest amount of overall warming is shown by EM15 (2.8K), while the largest is EM04 (4.3 K). All models show less warming in winter (December – February, DJF) with a range of 1.8K – 3.6K, than in summer (June – August, JJA) with a range of 3.5K – 5.1K. The spring (March – May, MAM) warms by 2.1 – 3.7 K and the autumn (September – November, SON) has the largest warming with a range of 3.6 – 5.2 K. Five of

the ensemble members show an increase in annual precipitation, with the largest increase being EMs 07 and 04 (5.3%), while seven show a decrease with the largest being EM01 (-6.6%). All models show an increase in DJF precipitation (7.6 – 29.0 %) and a decrease in JJA precipitation (-39.8 – -5.3 %). There is a mixed signal in SON with some decreasing and some increasing (-17.1 – 5.3 %), as well as in MAM (-15.0 – 14.3 %).

To span the range of the ensemble, we chose:

**EM01** This is the default parameterisation of the climate model, and is in the middle of the range of temperature increases, although it has one of the smallest increases in MAM temperature. It has the largest decrease in annual precipitation, as it has the smallest increase in DJF and the largest decrease in SON precipitation. It has a very small decrease in MAM precpitation and one of the smaller decreases in JJA.

**EM04** This has the largest increase in annual mean, MAM and SON temperature, and is near to the top of the range of DJF and JJA temperature increase. It has the second largest increase in annual precipitation, as it has one of the largest increases in DJF precipitation, a very small increase in MAM and SON precipitation and the smallest decrease in JJA.

**EM06** This is in the middle of the range of annual and seasonal warming. It has a moderate decrease in annual precipitation, which is due to a small increase in DJF precipitation combined with the largest decrease in JJA precipitation. It has the second largest increase in MAM precipitation and a small decrease in SON.

**EM15** This has the smallest increase in annual and DJF temperature and one of the lowest increases in MAM, JJA and SON temperature. It has very little change in annual and MAM precipitation, a moderate increase in DJF and decrease in JJA, and it has the second largest increase in SON precipitation.

This selection was made based on air temperature and precipitation as key drivers of change in the UK. However, focussing on these two variables does not necessarily ensure that the ranges of other variables are fully captured (McSweeney et al., 2014), particularly those for which changes are not driven by global temperature change (Hayashi and Shiogama, 2022). It is likely that a larger subset would be required to fully represent the uncertainties in multiple variables across seasons (Ito et al., 2020; McSweeney and Jones, 2016). Therefore this is a pragmatic selection that balances the requirements of impacts modelling against storage and processing limitations.

## 3.2 Downscaling

In order to create the 1 km resolution variables, we downscaled the 12 km resolution UKCP18 RCM-PPE variables to 1 km using an updated version of the CHESS downscaling methodology, which was originally used to downscale meteorological variables reported by MORECS (Hough et al., 1995) from the native resolution (40 km) to 1 km. The CHESS method uses physical and empirical relationships to add topographic variation to the interpolation between resolutions. However, there are differences between the variables available in MORECS and UKCP18, which motivated updates to the downscaling algorithms for this dataset. We have thus modified the CHESS method to enable processing of the UKCP18 RCM-PPE data. The code was implemented in Fortran and makes use of NAG routines (The Numerical Algorithms Group, 2011). The UKCP18 RCM-PPE

variables were adjusted to sea-level then interpolated from 12 km to 1 km resolution using a bicubic spline through the 12 km grid points, calculated using NAG routine `e01daf` (The Numerical Algorithms Group, 2011). The interpolated 1 km variables were then adjusted to the 1 km grid box elevation. The downscaling procedure is shown in Fig. 2. We summarise the procedure in this Section, and the full details are in Appendix A.

We downscaled the near-surface air temperature (daily mean, $T_a$ (K), minumum, $T_n$ (K), and maximum, $T_x$ (K)) by in-
terpolating and using a constant physical lapse rate of $\Gamma_T = -0.006$ C m$^{-1}$ to adjust for elevation (Hough et al., 1995). We calculated daily temperature range (DTR; $\Delta_T$ (K)) as the difference between daily maximum and minimum air temperature. We calculated the surface air pressure $p_*$ (Pa) by interpolating UKCP18 RCM-PPE 12 km sea level air pressure, then adjusting to grid box elevation using the hypsometric equation, which also involves the downscaled air temperature (Shuttleworth, 2012). We assumed the relative humidity was constant with elevation (cf Hough et al., 1995), so we simply interpolated it with no
adjustments. We calculated the specific humidity calculated as a function of downscaled relative humidity, air temperature and air pressure. We calculated the downwelling SW radiation by interpolating UKCP18 RCM-PPE 12 km net SW radiation, then adjusting for the slope and aspect of the 1 km grid box. In order to do this we also required the cloud area fraction, which we interpolated from UKCP18 RCM-PPE 12 km to 1 km. We then converted net to downwelling SW radiation using GlobAlbedo (European Space Agency, 2011). For physical consistency, we calculated the downwelling longwave (LW) radiation from the
downscaled air temperature and specific humidity, as well as the downscaled cloud area fraction. We scaled the precipitation using the ratio of the 1 km resolution empirical dataset of SAAR (Spackman, 1993) to its mean value at 12 km resolution. We interpolated the wind speed, then applied an offset based on the difference between the 1km ETSU mean wind speed dataset (Burch and Ravenscroft, 1992; Newton and Burch, 1985) and its mean value at 12 km resolution.

### 3.3   Bias correction

We used the CHESS-met observation-based climate dataset (Robinson et al., 2020a) as our reference data-led set of measurements, against which we compare the downscaled CHESS-SCAPE gridded estimates of meteorological conditions. Although CHESS-met is at the target 1 km resolution, we did not use it as a reference dataset for the downscaling as it was also produced by downscaling a coarser resolution dataset (MORECS (Robinson et al., 2017)) as well as interpolating station data (precipitation only (Keller et al., 2015)). In order to allow for differences between the input datasets we applied a modified version of
the CHESS methodology to create CHESS-SCAPE, and use CHESS-met as a reference for the bias-correction separately.

There are many options for bias-correction of GCM and RCM outputs (Teutschbein and Seibert, 2012; Watanabe et al., 2012; Jakob Themeßl et al., 2011), including methods which correct the mean or mean and variance such as linear scaling (Lenderink et al., 2007), methods which correct other statistical properties such as local intensity scaling (LOCI; Schmidli et al., 2006), methods that apply adjustments to the distributions of the data in either parametric or nonparametric ways (Gudmundsson et al.,
2012; Teutschbein and Seibert, 2012), and methods that apply to multiple timescales such as nested bias correction (Johnson and Sharma, 2012). In general, the CHESS-SCAPE data is found to be robust in its ability to reproduce the average features of the CHESS-met data, and so our approach to bias correction is parsimonious. For many of the CHESS-SCAPE variables it was sufficient to apply simple linear scaling, with an additive correction used for air temperature and specific humidity and

a multiplicative correction used for precipitation and wind speed (where subtracting a mean offset has the potential to make

some of the values negative, and so unphysical). In this case, the statistical distribution of the CHESS-SCAPE variables were very similar to that of the CHESS-met variables, so we did not find it necessary to perform higher order corrections. However, the distributions of the radiation variables were different enough that we required a form of parametric distribution mapping, in order to adjust the CHESS-SCAPE distributions to match those of the CHESS-met radiation variables.

For all variables, we calculated bias correction factors for each season (DJF, MAM, JJA and SON) and each 1 km grid

box by comparing the seasonal means of the CHESS-met data with the CHESS-SCAPE ensemble members for the overlap period 1980-2015. The CHESS-met dataset contains fewer land points than the CHESS-SCAPE data (notably, NI is absent in CHESS-met). As the bias correction requires the CHESS-met data, the resultant bias-corrected CHESS-SCAPE data also has fewer points, corresponding to the CHESS-met land points.

For air temperature and specific humidity we calculated the seasonal offsets $\mu_T$ and $\mu_q$ respectively as the difference between

CHESS-SCAPE and CHESS-met, and removed these offsets from the CHESS-SCAPE data. For precipitation and wind speed we calculated seasonal scaling factors $\mu_P$ and $\mu_u$ respectively as the ratio of CHESS-SCAPE to CHESS-met, and multiplied the CHESS-SCAPE data by these factors. For downwelling LW radiation we calculated the seasonal parameter $\mu_L$ to scale the values relative to an upper threshold of $400\,\mathrm{W\,m^{-2}}$, which chosen as the upper limit of the historical range of the LW radiation. For downwelling SW radiation we required two parameters, $\mu_{S,1}$ to scale the CHESS-SCAPE value when it is smaller than its

local seasonal mean, and $\mu_{S,2}$ to normalise against the local seasonal maximum when it is larger than the local seasonal mean. It was not possible to directly bias correct the daily minimum and maximum air temperatures as these variables do not exist in CHESS-met. Therefore we applied the bias-correction derived from the daily mean air temperature, $\mu_T$. This means that we were not able to correct for biases in the daily temperature range.

Maps of the temperature offset, $\mu_T$, and precipitation scaling, $\mu_P$, for each season and ensemble member are shown in

Figs. 3 and 4. Maps of the other bias-correction parameters can be seen in the Supplement. Timeseries of the 20-year seasonal mean values are shown in Figs. 5 and 6. In general, EM15 is warmer than the CHESS-met historical data by around 1 K (except MAM where it is 0.3 K cooler), while the other EMs are overall cooler by up to 1.5 K. In all ensemble members, there is an area in the south-east, which corresponds to the Greater London conurbation, which is warmer in the downscaled CHESS-SCAPE than in CHESS-met, even when CHESS-SCAPE is overall cooler. This is partly due to known urban calibration issues in

the UKCP18 RCM-PPE configuration, which are likely to be resolved in the CPM-PPE configuration (Fosser et al., 2019). All ensemble members are wetter in the in CHESS-SCAPE than in CHESS-met over much of England and low-lying regions in DJF and MAM, but are too dry in the highlands of Scotland at the same time. This leads to an overall wet bias in DJF and MAM, leading to a correction of -26% – -14% in DJF and -25% – 18% in MAM. All ensemble members except EM06 also have a small overall wet bias in JJA, which is due to a wet bias in the north-west of Scotland. All ensemble members

have very little bias in the precipitation in SON. For EM01, 14 and 06, the specific humidity required little bias-correction in DJF and SON, and was too low in MAM and JJA so was increased by 5% – 9% in those seasons. EM15 had too high specific humidity in DJF, JJA and SON, so the bias-correction reduced it by -6% – -5%, and required little bias correction in MAM. The downwelling LW radiation is slightly higher than CHESS-met in all four ensemble members, so the bias-correction reduced

it by between -13 W m$^{-2}$ and -2.5 W m$^{-2}$, which is a decrease of only a few percent. The downwelling SW radiation was very similar to CHESS-met in DJF for EM01, EM04 and EM15, but was too low in DJF in EM06, which required a correction of 9%. The bias correction increased downwelling SW radiation in MAM in EM01, 04 and 06 by between 6% – 11%, and decreased it in JJA by 3% – 12% for all ensemble members. There was a small decrease in SON for EM01, 04 and 06 and a larger decrease of around 10% in EM15. The wind speed was too high in EM01, 04 and 15 in all seasons, so was reduced by between 3% – 10%, and too low in EM06 so was increased by up to 6%.

### 3.3.1 Other variables

In order to ensure that the variables are physically consistent, we calculated the relative humidity from the bias-corrected specific humidity, bias-corrected air temperature and the downscaled-only surface air pressure, using the inverse of Eqs. A9 and A8 (in Appendix B). The combination of bias-correction of air temperature and specific humidity resulted in very small changes to the relative humidity of each ensemble member. The largest decrease was -4% in EM04 in DJF, the largest increase was 4% in EM06 in JJA. Other seasons were only changed by around 1%.

As the CHESS-met dataset only includes a low resolution reanalysis mean-monthly climatology of surface air pressure, and a low spatial and temporal resolution daily temperature range variable, it was not possible to use either to bias-correct the surface air pressure or daily temperature range of CHESS-SCAPE.

## 3.4 Alternative climate scenarios

Under the assumption of linearity in the climate system, changes to local climate variables are proportional to the change in global mean air temperature, no matter the trajectory taken (Mitchell, 2003). Although there are some processes that can cause changes to variables that are non-linear with temperature at a global scale (Mitchell et al., 2016), the linearity assumption has been well tested at a local and regional scale (eg Zelazowski et al., 2018). Using this assumption, new climate change scenarios can be derived from existing GCM or RCM outputs, using techniques such as time shifting (directly substituting years with the required global temperature change, eg Herger et al. (2015); Schleussner et al. (2016)) or pattern scaling (using derived relationships between global temperature change and local climate variables, eg Huntingford and Cox (2000); James et al. (2017)). Similar methods were used to create the derived scenarios in UKCP18 (Gohar et al., 2018).

For this study, we made use of the downscaled RCP8.5 scenario to derive scenarios corresponding to the other three RCPs (2.6, 4.5 and 6.0), using a combination of time shifting and pattern scaling. The procedure was:

1. we defined a target global mean air temperature trajectory;

2. we time shifted the data by finding the year of the CMIP5 RCP8.5 scenario for which the global mean air temperature is within a threshold difference from the target CMIP5 trajectory;

3. we applied pattern scaling to the resulting time shifted trajectory based on the difference between the global mean air temperature of the time shifted RCP8.5 and target CMIP5 trajectories for each year.

This was applied to all years starting in December 2010, with the historical period 1980 – 2010 being identical for all scenarios.

### 3.4.1 Time shifting

In order to define target global warming trajectories, we required GCM output for the three alternative RCPs. Since many entries to CMIP5 provided model output for these scenarios, we selected one proxy CMIP5 model for each of the CHESS-SCAPE ensemble members. We selected the CMIP5 models by comparing the annual global mean air temperature anomalies of the
UKCP18 GCM-PPE RCP8.5 ensemble with the subset of the CMIP5 database for which all RCPs were available. We defined the closest match to be the model with the highest Kling-Gupta efficiency (KGE) for global annual mean air temperature. If more than one ensemble member was best represented by the same CMIP5 model, then we chose the next nearest CMIP5 match instead. The KGE values were between 0.78 and 0.93 (Table 5). Three of the four UKCP18 ensemble members used were found to match best with ensemble members of the Met Office Hadley Centre HadGEM2-ES model (Collins et al., 2011).
This is unsurprising as the HadGEM2-ES and HadGEM3-GC3.05 are closely related configurations of the same climate model.

In order to define the new time-shifted scenarios, for each year of each CMIP5 target scenario, we randomly selected a year of the RCP8.5 scenario which had a global annual mean temperature anomaly within a threshold of 0.5 K difference from the target warming, ensuring that there were no repeats within any 20 year period. After these initial time shifted scenarios were found, we applied pattern scaling to bring the derived scenarios closer to the target.

### 3.4.2 Pattern scaling

We found the pattern of change in meteorological variables by calculating the linear regression of the seasonal mean anomaly (or relative anomaly) of each variable to the global annual mean air temperature given by the corresponding ensemble member of the UKCP18 GCM-PPE (Huntingford and Cox, 2000). We then adjusted the time shifted scenarios by multiplying these seasonal patterns by the difference between the global mean air temperature of the time shifted and target CMIP5 scenarios,
then using this to scale the daily timeseries for each season. We carried out this procedure for both the bias-corrected and the downscaled-only variables.

The seasonal patterns of air temperature and precipitation can be seen in Figs. 7 and 8, and the other variables are in the Supplementary Information. The UK air temperature change is overall higher than the global annual mean air temperature change in JJA and SON, and less in DJF and MAM. Overall UK precipitation decreases with increasing air temperature in
JJA and increases in DJF. However there are some regions of Scotland and eastern England where JJA increases with air temperature in EM01 and EM04, and in Scotland and NI where DJF precipitation decreases with air temperature in EM01 and EM06. This is related to differences in the change in global circulation patterns across the ensemble, in particular the North Atlantic Oscillation (NAO Murphy et al., 2018), that have a strong impact on UK rainfall (Kendon et al., 2021; Trigo et al., 2002). The picture is mixed in MAM and SON, with each ensemble member showing a different pattern of increasing and
decreasing precipitation.

### 3.4.3 Derived variables

To ensure that the variables are physically consistent, we did not apply time shifting and pattern scaling to relative humidity. Instead we calculated the relative humidity from the specific humidity, air temperature and surface air pressure.

For the variables without bias-correction, we calculated the gradient of daily minimum and maximum air temperature with respect to global mean air temperature and applied the pattern scaling directly. For the bias-corrected variables we instead applied the pattern scaling that was calculated from the bias-corrected daily mean air temperature.

## 4 Dataset description

The full CHESS-SCAPE dataset comprises four ensemble members: 01, 04, 06 and 15. Each ensemble member has four scenarios: RCP2.6, RCP4.5, RCP6.0 and RCP8.5. And each scenario is provided as a downscaled-only and a downscaled-bias-corrected version. The downscaled-only scenarios contain 11 variables (see Table 2). The bias-corrected scenarios do not contain daily temperature range (as it is identical to the downscaled-only version) or surface air pressure (as insufficient data were available for bias correction). The non-bias-corrected daily temperature range and surface air pressure may be used instead alongside the bias-corrected versions of the other variables.

The data are published as CF-compliant netCDF files. The spatial grid is defined on the OSGB grid (EPSG, a). The spatial extent is from (0, 0) to (656000, 1057000) in OSGB eastings and northings. The data are provided at a daily timestep. The dataset also includes several time averages: monthly, seasonal and annual means, as well as "time-slices," which are 20-year mean monthly climatologies at 10 year intervals.

Maps of air temperature and precipitation for the first timeslice (1980-2000) of the bias-corrected variables are shown in Figs. 9 and 10. The other variables, and the downscaled-only variables are shown in the Supplementary Information. These are consistent with the historical CHESS-met dataset (Robinson et al., 2020a), and show a gradient from cooler, wetter conditions in the north-west to warmer, drier conditions in the south-east. While the bias-correction corrects differences between the climate model and the historical baseline, the climate model still replicates this gradient well and it can be seen in the downscaled-only variables as well. Maps of the air temperature and precipitation anomaly in 2060-2080, compared to the 1980-2000 baseline, is shown in Figs. 11 and 12. The UK mean annual and seasonal air temperature anomalies in 2060-2080 are given in Table 6, and the mean precipitation anomalies are in Table 7. The CHESS-SCAPE RCP8.5 scenarios preserve the change in air temperature seen in UKCP18 RCM-PPE, with an increase in annual mean air temperature between 2.8 – 4.3 K between 1980–2000 and 2060–2080. The other warming scenarios have lower air temperature increases, with RCP2.6 the lowest at 1.0-1.9 K and RCP6.0 the highest at 1.5 – 3.4 K. Although RCP4.5 and RCP6.0 have similar warming at the end of the projections, as they are following differing trajectories where RCP4.5 has a peak in emissions at 2040 followed by a decline to 2100, whereas RCP6.0 emissions rise slightly more slowly, but continue rising to 2100. Since UKCP18 and therefore CHESS-SCAPE end in 2080, the two scenarios remain quite similar in the final timeslice (2060 – 2080). The temperature change is the same in the downscaled-only and the downscaled and bias-corrected data.

Again, the CHESS-SCAPE RCP8.5 scenarios preserve the precipitation change from UKCP18, with small annual changes (between -6 – 6 %), moderate to large increases in DJF precipitation (9 – 22%) and large decreases in JJA precipitation (-39 – -14 %). The precipitation changes in the alternative RCPs are quite variable. The decrease in JJA precipitation is broadly proportional to the global air temperature change, with the largest decreases apart from RCP8.5 seen in RCP6.0 (-26 – -5 %), except EM01 which has the largest decrease in JJA precipitation in RCP4.5 (-11%). However, DJF precipitation changes are more variable, and not proportional to the warming scenario. The largest decrease (-9%) and largest increase (15%) are both in RCP2.6. However, this represents the high inter-annual variability of UK precipitation, particularly in DJF (Kendon et al., 2021). After bias correction, the percentage change in precipitation is slightly altered, but the difference is small and the same patterns of change are preserved. The spatial pattern of precipitation change is complex, with some scenarios showing a different sign of change between north and south in particular. For all RCPs for EM01, there is an increase in DJF precipitation over much of England, but a decrease is seen in the north of Scotland. Similar is true for EM06 for all but RCP2.6.

Overall the specific humidity increases by the end of the projections, with the smallest relative increases in RCP2.6 and the largest in RCP8.5. The increases tend to be slightly larger in the south and east than the north and west. However, because the temperature also increases, the relative humidity actually decreases in MAM, JJA and SON, with the largest decreases of 7% in JJA. The DJF relative humidity shows very little change. The decrease in relative humidity is larger in the south and east than in the north and west. Very little change is seen in the surface air pressure in any of the scenarios. The change in wind speed is mixed for DJF and MAM, with some ensemble members showing an increase and some a decrease, ranging between -3 m s$^{-1}$ and 3 m s$^{-1}$, and little correlation with RCP. For JJA and SON there is a consistent decrease across the ensemble, for RCP6.0 and 8.5, but the decrease is still small at -0.4 m s$^{-1}$ – 0.1 m s$^{-1}$. The other RCPs show some increase and decrease. Although there is spatial variation in the magnitude of the change, the sign of the change tends to be consistent across the country. The downwelling SW is projected to increase in all seasons, with the largest increase in JJA and with larger increases associated with more extreme scenarios. This is due to a projected decrease in cloud cover, particularly in JJA. The increase in DJF tends to be higher in Scotland, while in MAM and JJA it is higher in England. The downwelling LW radiation also increases through the projections, with a larger increase associated with scenarios with larger amounts of warming, but overall the change is relatively small, with increases of 2 W m$^{-2}$ – 10 W m$^{-2}$ in RCP2.6 and 7 W m$^{-2}$ – 21 W m$^{-2}$ in RCP8.5. Although increasing temperature will drive an increase in downwelling LW, it is likely to be mitigated by the projected decrease in cloud cover. The increase in downwelling LW tends to be larger in the north than the south.

## 4.1 Extremes

In order to evaluate the representation of temperature extremes in the CHESS-SCAPE dataset, we calculated the 90th percentile of daily mean air temperature, $T_{a,90}^{\mathrm{C}}$ (°C) the first twenty years of the dataset (1980–2000). We compared the 1980–2000 values to the 90th percentile of daily mean air temperature from the CHESS-met dataset for 1981-2000, $T_{a,90}^{\mathrm{m}}$ (°C). The upper panels of show the GB mean of the 90th percentiles of daily mean air temperature for each pixel of CHESS-SCAPE plotted against the corresponding pixel of CHESS-met for both the downscaled-only and the bias-corrected ensemble members. Before bias-correction there is a good correspondence between the CHESS-SCAPE and CHESS-met 90th percentiles, with gradients of

between 0.92 – 1.02, mean bias error (MBE) of between -0.99 – 0.85 °C and $r^2$ values of 0.98 for all ensemble members. After bias correction the $r^2$ is increased to between 0.99 – 1.00 and the MBE is improved to between -0.51 – -0.10 °C for three ensemble members, although for EM01 the MBE is slightly increased to to 0.26 °C. The gradient of the fit is slightly lower for all ensemble members at 0.91–0.95 as the lower value 90th percentiles tend to be slightly higher than CHESS-met. The lower panels show violin plots of the distributions of the 90th percentiles of daily mean air temperature for CHESS-met and CHESS-SCAPE and of daily maximum air temperature for CHESS-SCAPE. This shows that although the bias-correction is only applied as a mean bias change, it also adjusts the distribution of the 90th percentiles of daily mean air temperature to better match CHESS-met, and shows that it has a similar effect on the 90th percentiles of daily maximum air temperature.

We then calculated the 90th percentile of daily mean air temperature for 2060–2080, and the 90th percentile of daily maximum air temperature $T^{\mathrm{C}}_{x,90}$ (°C) for 1980–2000 and 2060–2080 to investigate the change over the projections. The projections show that the 90th percentiles of daily maximum air temperature will increase by 0.6 – 3.5 K for RCP2.6 and 2.2 – 7.6 K for RCP8.5 (Fig. 14). For each ensemble member and RCP increase in daily maximum air temperature is larger than the increase in the corresponding daily mean air temperature. This difference is larger for the more extreme warming scenarios, with on average an increase of 1.14 K in daily maximum air temperature for every 1 K increase in daily mean air temperature. This is consistent with other studies indicating an expected increase in the variability of extreme temperatures in the UKCP18 RCM-PPE (Kennedy-Asser et al., 2021). In the final 20 years of the projections, the temperature increases mean that in RCP2.6 13 – 30 % of days exceed the 1980–2000 90th percentile, while in RCP8.5 24 – 46 % of days exceed the 1980–2000 90th percentile. The number of days exceeding this threshold tends to be larger in the south than the north (Fig. 15), and the increase is particularly large in EM04 where the majority of the country has more than 35 % of days exceeding the 1980–2000 90th percentile. In 2060–2080 after bias-correction between 6 and 15 years have at least one day with daily maximum air temperature exceeding 40 °C somewhere, a threshold which has thus far only been exceeded in one year since records began (Kendon et al., 2023).

We also calculated 10-year return levels of annual maximum precipitation, $P_{x,10\mathrm{y}}$ (mm/day). In order to obtain robust statistics, we used thirty year periods, so calculated the 10-year return levels for the first thirty years (1980–2010) and the final thirty years (2050–2080). Figure 16 shows maps of the 10-year return levels for 1980–2010 for the downscaled-only and the bias-corrected ensemble members (note that this is identical for all RCPs as the scenarios only differ from December 2010 onwards). These range from 28 – 223 kg m$^{-2}$ d$^{-1}$ for the downscaled-only data and are reduced to 23 – 204 kg m$^{-2}$ d$^{-1}$ after bias correction. The spatial distribution follows the rainfall patterns in Fig. 6 with a strong north-west to south-east gradient. Before bias-correction the $r^2$ of a linear fit of the CHESS-SCAPE to CHESS-met is 10-year return levels 0.34 – 0.54, while after bias-correction it rises to 0.58 – 0.74. Before bias-correction, CHESS-SCAPE tends to underestimate the return levels in the highlands of Scotland, but there are large areas where the return levels are overestimated, especially coastal areas, due to the overall wet bias in UKCP18 RCM-PPE (maps of the difference in 10-year return levels between CHESS-SCAPE and CHESS-met are show in the Supplementary Information). After bias-correction, although the largest differences are reduced, CHESS-SCAPE still overestimates the 10-year return levels in some areas, particularly the south-west. But the bias-corrected CHESS-SCAPE underestimates the return levels across more of the country, reflecting the fact that it is difficult for CHESS-

SCAPE to represent the most extreme rainfall because of the lack of explicit convection in the input UKCP18 RCM-PPE runs. However the bias-correction does afford an improvement over the downscaled-only ensemble members.

The change in 10-year return levels of annual maximum precipitation between 1980–2010 and 2050–2080 for the downscaled-only data are shown in Fig. 17. By the end of the century, the return levels almost double in some places, indicating increasing precipitation intensity. However, overall the picture is mixed, with the 10-year return level falling in 27 – 35 % of the land area for downscaled-only RCP2.6, 15 – 39 % RCP4.5, 7 – 38 % for RCP6.0 and 3 – 29 % for RCP8.5. For example, EM15 shows a band of decreasing 10-year return level across England in all scenarios, which correspond to areas where the 10-year return levels are overestimated compared to CHESS-met in the historical period. This reflects the variability introduced by the response of circulation patterns to global temperature increases (Zappa and Shepherd, 2017).

## 4.2 Comparison with other UKCP18 projections

Comparison withe the other UKCP18 projections showed that CHESS-SCAPE preserved and mirrored the spatial and temporal patterns of these datasets. Figures 18 and 19 show the 20-year mean seasonal anomalies of the CHESS-SCAPE downscaled-only variables, averaged over the UK for each ensemble member, RCP and season. For each RCP and season we plot the 5th, 50th and 95th percentiles of the UKCP18 probablisitic projections (Met Office Hadley Centre, 2018b). For RCP8.5 we also plot the twenty-year timeslice seasonal means of the three UKCP18 CPM-PPE timeslices: 1980-2000, 2020-2040 and 2060-2080 (Met Office Hadley Centre, 2018a). The same plots for the other variables are in the Supplementary Information.

The climate in the dataset is determined by the underlying UKCP18 RCM-PPE projections, so the UK mean climate change is preserved after downscaling and bias-correcting. However, it is useful to compare to other strands of UKCP18: the UKCP18 CPM-PPE projections, which are of a similar spatial resolution, and the UKCP18 probabilistic projections, which are lower resolution, but provide projections for all four RCPs. For RCP8.5 the CHESS-SCAPE air temperatures are consistent with UKCP18 CPM-PPE, given that these are largely determined by the GCM in which the CPM and the RCM-PPE are nested. Both CHESS-SCAPE and UKCP18 CPM-PPE are consistent with the upper half of the range of the UKCP18 probabilistic estimates, although EM04 lies just above the 95th percentile for MAM and SON, and EM15 is below the median for DJF. The changes to precipitation in CHESS-SCAPE tend to be somewhat smaller than in the UKCP18 CPM-PPE projections. This is likely to be due to the better resolution of rainfall with the CPM. In particular the free convection will better resolve JJA storms. Although the CHESS-SCAPE precipitation changes are smaller, they are consistent in sign with UKCP18 CPM-PPE, and are consistent with the UKCP18 probabilistic projections for all RCPs, including RCP8.5.

The increase in specific humidity and decrease with relative humidity is consistent with the UKCP18 CPM-PPE and probabilistic projections. The very small change in surface air pressure in any of the scenarios is consistent with the UKCP18 CPM-PPE timeslices and the UKCP18 probabilistic scenarios, the latter of which have a range which includes zero. The changes in wind speed are consistent with the UKCP18 CPM-PPE results, but there is no probabilistic projection for wind speed. The change in the downwelling SW radiation is high compared to the UKCP18 probabilistic projections, which is because the net SW in the UKCP18 RCM-PPE projections is also high. Downwelling SW is not available for the UKCP18 CPM-PPE or projections, but a comparison of net SW shows that the CPM-PPE projections are at the top of the range of or

higher than the probabilistic projections. Both the probabilistic and CPM-PPE projections show a projected decrease in cloud cover, particularly in RCP8.5, leading to these changes in downwelling SW radiation, but the climate model (RCM-PPE and CPM-PPE) has a stronger response in SW radiation than the probabilistic projections. There is no downwelling LW in the UKCP18 CPM-PPE or probabilistic projections, but they show little change in the net LW. Although the CPM-PPE projections do show a decrease in JJA net LW, it is small and of a similar magnitude to the CHESS-SCAPE projected downwelling LW change.

The CHESS-SCAPE dataset is consistent with the projections provided by these strands of UKCP18, with the enhancements of high spatial resolution, alternative warming scenarios and bias-correction.

## 5 Conclusions

Using a combination of RCM output and physical and empirical downscaling, we have produced a comprehensive set of high-resolution climate projections for the UK. CHESS-SCAPE is derived from one strand of UKCP18 and is consistent with the climate projections in the other strands. The CHESS-SCAPE climate change trajectories are consistent with moderate to high end of the UKCP18 probabilistic projections for all RCPs. This reflects the underlying nested GCM and RCM, which have relatively high equilibrium climate sensitivity. The climate change trajectory for RCP8.5 is consistent with the UKCP18 CPM projections, although CHESS-SCAPE has somewhat less extreme changes in precipitation. The bias-corrected CHESS-SCAPE are consistent with the CHESS-met observations in the historical period.

CHESS-SCAPE enables modelling of the land surface and climate-change impacts at an appropriate resolution, and enables exploration of multiple warming scenarios and of model uncertainty through the four-member ensemble. The bias-correction allows for modelling that is consistent with the historical period to be take through to the future seamlessly, alongside provision of the downscaled-only data that are consistent with the raw climate model output. The high spatial and temporal resoution, and the addition of alternative warming scenarios and bias-correction makes CHESS-SCAPE a comprehensive climate change dataset, which can be used for a wide variety of impacts modelling for the future.

## 6 Data availability

The CHESS-SCAPE data are available for download from the NERC EDS Centre for Environmental Data Analysis: http://dx.doi.org/10.5285/8194b416cbee482b89e0dfbe17c5786c (Robinson et al., 2022). The data are provided under the terms of the Open Government Licence v3.0 (Open Government Licence, 2022).

## Appendix A: Downscaling methods

In this section, the 12 km RCM-PPE data are denoted by the superscript "U", while the downscaled 1 km CHESS-SCAPE data are denoted by the superscript "C".

## A1    Air temperature: mean, minimum, maximum and range

We first reduced the daily mean air temperature ($T_a^U$, °C) to mean sea level, using a constant lapse rate of $\Gamma_T = -0.006\ \mathrm{C\ m^{-1}}$ and the lapse rate equation for converting temperature at elevation $z_1$ to elevation $z_2$:

$$T_a^U(z_2) = T_a^U(z_1) + (z_2 - z_1)\Gamma_T. \tag{A1}$$

To convert to sea level, $z_1$ is the RCM grid box elevation and $z_2$ is zero. We then interpoalted the sea-level air temperature to 1 km, before adjusting it to the elevation of the 1 km grid by applying Eq. A1 to the downscaled sea-level air temperature, with $z_1 = 0$ and $z_2$ equal to the 1 km grid box elevation. Finally, we converted it to the required units to give $T_a^C$ (K).

We applied the same routine was to daily minimum air temperature ($T_n^C$, K) and daily maximum air temperature ($T_x^C$, K). Finally, we found the daily temperature range (DTR; $\Delta_T^C$, K) as the difference

$$\Delta_T^C = T_x^C - T_n^C. \tag{A2}$$

However, because the interpolation was unconstrained, it sometimes resulted in the interpolated minimum air temperature being larger than the interpolated maximum, or the interpolated mean being outside of the range given by the interpolated minimum and maximum. Thus, we included an adjustment to check that all three temperatures were consistent.

- If the minimum was larger than the maximum, they were adjusted so that

$$
\begin{aligned}
T_n^C &= T_a^C - \frac{\Delta_{\min}^C}{2} & \text{(A3)}\\
T_x^C &= T_a^C + \frac{\Delta_{\min}^C}{2}, & \text{(A4)}
\end{aligned}
$$

  where $\Delta_{\min} = 0.05$ K is the minimum allowed DTR. This limit was set based on inspection of the UKCP18 RCM-PPE 12 km data.

- If the mean was outside of the range given by the minimum and maximum, the calculated DTR, $\Delta_T$, was preserved, but the minumum and maximum were adjusted so that the mean lies between them

$$
\begin{aligned}
T_n^C &= T_a^C - \frac{\Delta_T^C}{2} & \text{(A5)}\\
T_x^C &= T_a^C + \frac{\Delta_T^C}{2}. & \text{(A6)}
\end{aligned}
$$

## A2    Air pressure

Daily mean surface air pressure at 1 km resolution, $p_*^C$ (Pa), is obtained from the UKCP18 RCM-PPE sea-level air pressure, $p_{\mathrm{sl}}^U$ (hPa). The sea-level air pressure was interpolated to 1 km using a bicubic spline, then multiplied by 100 to convert to Pa. This was then adjusted to the CHESS-SCAPE elevation using the integral of the hypsometric equation, such that

$$p_*(z_2) = p_*(z_1)\left(\frac{T_a(z_1)}{T_a(z_2)}\right)^{\frac{gM_a}{R_a\Gamma_T}}, \tag{A7}$$

where $g = 9.81$ m s$^{-2}$ is acceleration due to gravity, $M_a = 0.0289644$ kg mol$^{-1}$ is the average molecular mass of air and $R_a = 8.31432$ N m mol$^{-1}$ K$^{-1}$ is the ideal gas constant for air (Shuttleworth, 2012). In this case, $z_1$ was zero and $z_2$ was the 1 km grid box elevation.

## A3 Relative humidity

This was assumed to be constant with elevation, so we interpolated the UKCP18 RCM-PPE relative humidity, $R^{\mathrm{U}}$ (%), to 1 km resolution, $R^{\mathrm{C}}$ (%), with no sub-grid corrections applied.

## A4 Specific humidity

We calculated the daily mean specific humidity, $q_a^{\mathrm{C}}$ (kg kg$^{-1}$) by converting the CHESS-SCAPE 1 km relative humidity, $R^{\mathrm{C}}$, to specific humidity, using

$$q_a^{\mathrm{C}} = \frac{\epsilon e_a^{\mathrm{C}}}{\frac{p_*^{\mathrm{C}}}{100} - (1 - \epsilon) e_a^{\mathrm{C}}}, \tag{A8}$$

where $\epsilon = 0.622$ is the mass ratio of water to dry air (Gill, 1982), where $e_a^{\mathrm{C}}$ (Pa) is the vapour pressure at 1 km resolution, given by

$$e_a^{\mathrm{C}} = \frac{R^{\mathrm{C}}}{100} \frac{e_s^{\mathrm{C}} p_*^{\mathrm{C}}}{p_*^{\mathrm{C}} + e_s^{\mathrm{C}} \left( \frac{R^{\mathrm{C}}}{100} - 1 \right)} \tag{A9}$$

and $e_s^{\mathrm{C}}$ (Pa) is the vapour pressure at saturation, which we calculated using the Richards (1971) empirical fit to air temperature


$$e_s^{\mathrm{C}} = p_{\mathrm{sp}} \exp \left( \sum_{i=1}^{4} a_i \left( 1 - \frac{T_{\mathrm{sp}}}{T_a^{\mathrm{C}}} \right)^i \right), \tag{A10}$$

where $p_{\mathrm{sp}} = 101325$ Pa is steam point pressure, $T_{\mathrm{sp}} = 373.15$ K is steam point temperature and $a = $ (13.3185, -1.9760, -0.6445, -0.1299) is a vector of empirical coefficients (Richards, 1971).

## A5 Downwelling shortwave radiation

We calculated the downwelling SW radiation, $S_d^{\mathrm{C}}$ (W m$^{-2}$) from the UKCP18 RCM-PPE net SW radiation, $S_n^{\mathrm{U}}$ (W m$^{-2}$). First we interpolated the net SW from 12 km to 1 km using a bicubic spline. We then adjusted this for the mean inclination and aspect of the 1 km grid box, which we calculated at 50 m resolution from the IHDTM and OSNI DTM using the methods of Horn (1981) and then aggregated to 1 km. We assumed the diffuse fraction of the radiation is unaffected by inclination and aspect, and we scaled the direct beam radiation by finding the ratio of the top of atmosphere radiation calculated for a
horizontal and an inclined plane (Allen et al., 2006). We assumed the diffuse fraction of the net SW radiation, $f_{\mathrm{diff}}$, is equal to cloud area fraction (Muneer and Munawwar, 2006). We therefore interpolated the UKCP18 RCM-PPE cloud area fraction ($C_f^{\mathrm{U}}$, %) to 1 km ($C_f^{\mathrm{C}}$, %) using a bicubic spline, and used this as the 1 km diffuse fraction of SW radiation.

We obtained the 1 km downwelling SW from 1 km net SW ($S_n^C$, W m$^{-2}$), using

$$S_d^C = \frac{S_n^C}{1 - \alpha} \tag{A11}$$

where $\alpha$ is the albedo of the surface. We calculated mean monthly values of albedo for each 1 km grid box using monthly climatological values of white-sky albedo ($\alpha_w$) and black-sky albedo ($\alpha_b$) from GlobAlbedo, combined using the 1 km cloud area fraction such that

$$\alpha = \left(1 - C_f^C\right) \alpha_b + C_f^C \alpha_s. \tag{A12}$$

## A6   Downwelling longwave radiation

We calculated downwelling LW radiation, $L_d^C$ (W m$^{-2}$) from 1 km air temperature, specific humidity and cloud area fraction, using the CHESS methodology (Robinson et al., 2017). We calculated the downwelling LW for clear-sky conditions using the method of Dilley and O'Brien (1998), such that

$$L_d^C = 59.38 + 113.7 \left(\frac{T_a^C}{T_{\text{ref}}}\right)^6 + 96.96 \sqrt{\frac{w^C}{w_{\text{ref}}}}, \tag{A13}$$

where $w$ (kg m$^{-2}$) is the precipitable water content of the atmosphere and $T_{\text{ref}}$ = 273.16 K and $w_{\text{ref}}$ = 25 kg m$^{-2}$ are reference

temperature and precipitable water respectively. We calculated the precipitable water, $w^C$, following Prata (1996),

$$w^C = 465 \frac{e_a^C}{T_a^C}, \tag{A14}$$

using air temperature and the vapour pressure from Eq. A9. We calculated the additional downwelling LW due to cloud cover from the cloud area fraction using the equations of Kimball et al. (1982), assuming a constant cloud base height of 1000 m (Robinson et al., 2017).

## A7   Precipitation

We did not interpolate the daily mean precipitation, $P^C$ (kg m$^{-2}$ s$^{-2}$), instead we found the 1 km grid box value using a scaling derived from the 1 km SAAR dataset (Spackman, 1993). The 1 km SAAR dataset for GB is defined on a grid that is offset by 500 m (half a grid box) from the CHESS-SCAPE grid, and the SAAR for NI is defined on the OSNI projection, so we first interpolated from the native SAAR grids to the CHESS-SCAPE grid. We aggregated these interpolated 1 km SAAR values,

$P_{\text{SAAR}}^{\text{1km}}$, to 12 km resolution, $P_{\text{SAAR}}^{\text{12km}}$. We then calculate the ratio of each 1 km SAAR value to the SAAR in the corresponding 12 km grid box. We used this as a constant scaling factor for the 12 km UKCP18 precipitation, $P^U$ (kg m$^{-2}$ d$^{-2}$) and converted to kg m$^{-2}$ s$^{-2}$, such that

$$P^C = \frac{P^U}{t_d} \frac{P_{\text{SAAR}}^{\text{1km}}}{P_{\text{SAAR}}^{\text{12km}}}, \tag{A15}$$

where $t_d$ = 86400 s is the number of seconds in a day.

## A8    Wind speed at 10 m

We calculated the daily mean wind speed at 10 m, $u_{10}^{\mathrm{C}}$ (m s$^{-1}$) by interpolating the UKCP18 12 km wind speed, $u_{10}^{\mathrm{U}}$ (m s$^{-1}$), using a bicubic spline. We then applied a correction based on the ETSU mean wind speed dataset (Burch and Ravenscroft, 1992; Newton and Burch, 1985). We aggregated the 1 km ETSU values ($u_{\mathrm{ETSU}}^{1\mathrm{km}}$, m s$^{-1}$) to 12 km ($u_{\mathrm{ETSU}}^{12\mathrm{km}}$, m s$^{-1}$), then calculated the difference between 1 km values and the corresponding 12 km grid box. We applied this difference as an offset to the 1 km interpolated wind speed ($u_{10}^{1\mathrm{km}}$), such that

$$u_{10}^{\mathrm{C}} = u_{10}^{1\mathrm{km}} + \left( u_{\mathrm{ETSU}}^{1\mathrm{km}} - u_{\mathrm{ETSU}}^{12\mathrm{km}} \right) \tag{A16}$$

## Appendix B:  Bias correction methods

In this section the downscaled-only CHESS-SCAPE data are denoted by superscript "C", while the downscaled and bias-corrected CHESS-SCAPE data are denoted by the superscript "BC". The reference dataset, CHESS-met, is denoted by the superscript "m".

We calculated bias correction factors for each location $i$, for each season $j$ (DJF, MAM, JJA and SON) and for each variable. In order to determine the number of parameters required for bias correction, we compared features of distributions of the daily values for each season in the common period of 1980-2015. If $k = 1 \dots n_y$ is the year, $s = 1 \dots 4$ denotes the season and $l = 1 \dots n_d$ is the day in each season, then the mean of variable $x(i,j,k,s,l)$ for location $(i,j)$ and season $s$ is given by where

$$\overline{x}(i,j,s) = \frac{1}{n_y n_d} \sum_{k=1}^{n_y} \sum_{l=1}^{n_d} x(i,j,k,s,l) \tag{B1}$$

where $n_y = 35$ is the number of years in the common period between the CHESS-SCAPE and CHESS-met data and $n_d$ is the number days in the season. In CHESS-SCAPE, which has a 360-day calendar, $n_d = 90$. In CHESS-met, which has a Gregorian calendar, $n_d$ depends on season. In DJF $n_d = 90$ (or 91 in leap years), in MAM and JJA $n_d = 92$ and in SON $n_d = 91$.

## B1    Air temperature

The variability of the 1.5m air temperature is very similar between the two datasets, but the seasonal means showed some differences. Therefore we calculated a simple offset, given by

$$\mu_T(i,j,s) = \overline{T_a^{\mathrm{C}}}(i,j,s) - \overline{T_a^{\mathrm{m}}}(i,j,s), \tag{B2}$$

where $T_a^{\mathrm{C}}$ is the air temperature in the reference period of the CHESS-SCAPE dataset and $T_a^{\mathrm{m}}$ is the air temperature in the CHESS-met dataset. We then applied the bias-correction to the whole period 1980-2080. Thus the bias-corrected CHESS-SCAPE air temperature, $T_a^{\mathrm{BC}}$ is given by

$$T_a^{\mathrm{BC}}(i,j,k,s,l) = T_a^{\mathrm{C}}(i,j,k,s,l) - \mu_T(i,j,s). \tag{B3}$$

## B2 Specific humidity

The specific humidity also only required a correction to the mean bias, so we calculated the offset of the 1.5 m specific humidity to be

$$\mu_q(i,j,s) = \overline{q_a^{\mathrm{C}}}(i,j,k,s,l) - \overline{q_a^{\mathrm{m}}}(i,j,k,s,l). \tag{B4}$$

The bias-corrected specific humidity is given by

$$q_a^{\mathrm{BC}}(i,j,k,s,l) = q_a^{\mathrm{C}}(i,j,k,s,l) - \mu_q(i,j,s). \tag{B5}$$

## B3 Precipitation

For precipitation, again the variability was very similar, so the only correction required was to remove the mean bias. Since removing an absolute difference could result in unphysical negative values, we instead calculated a scaling correction to be

$$\mu_P(i,j,s) = \frac{\overline{P^{\mathrm{C}}}(i,j,s)}{\overline{P^{\mathrm{m}}}(i,j,s)} \tag{B6}$$

The bias-corrected precipitation was thus given by

$$P^{\mathrm{BC}}(i,j,k,s,l) = \frac{P^{\mathrm{C}}(i,j,k,s,l)}{\mu_P(i,j,s)}. \tag{B7}$$

## B4 Wind speed

The wind speed also only required a correction to the mean bias, and a scaling correction was applied to avoid unphysical negative wind speeds, such that

$$\mu_u(i,j,s) = \frac{\overline{u^{\mathrm{C}}}(i,j,s)}{\overline{u^{\mathrm{m}}}(i,j,s)} \tag{B8}$$

The bias-corrected wind speed was thus given by

$$u^{\mathrm{BC}}(i,j,k,s,l) = \frac{u^{\mathrm{C}}(i,j,k,s,l)}{\mu_u(i,j,s)}. \tag{B9}$$

## B5 Downwelling longwave radiation

The radiation terms are more complex, and could be characterised by a simple offset or scaling. For downwelling LW radiation, inspection of the distributions showed a smaller difference between higher radiation values in CHESS-SCAPE and CHESS-met than between the lower values. The difference between the distributions was best characterised by the difference of the seasonal means from an upper threshold of 400 W m$^{-2}$. We defined the distributions relative to this upper threshold via a parameter $\mu_L$:

$$\mu_L(i,j,s) = \frac{\overline{L_d^{\mathrm{C}}}(i,j,s) - 400.0}{\overline{L_d^{\mathrm{m}}}(i,j,s) - 400.0}. \tag{B10}$$

The bias-corrected downwelling LW is then given by

$$L_d^{\mathrm{BC}}(i,j,k,s,l) = 400.0 + \frac{(L_d^{\mathrm{C}}(i,j,k,s,l) - 400.0)}{\mu_L(i,j,s)}. \tag{B11}$$

This has the effect of applying a larger correction to the lower CHESS-SCAPE values and a smaller correction to the larger values, and compressing or stretching the range of the distribution as necessary.

## B6  Downwelling shortwave radiation

Comparing the distributions of the downwelling SW radiation, showed that the values at the higher end of the distribution were similar between CHESS-SCAPE and CHESS-met, but the lower and middle parts of the distribution could be offset substantially. Therefore we apply a form of distribution mapping using two parameters, which adjust the lower and upper parts of the distrubution separately. The first parameter is a scaling of the distribution for when the CHESS-SCAPE value is smaller than its local and seasonal mean. This parameter is given by:

$$\mu_{S,1}(i,j,s) = \frac{\overline{S_d^{\mathrm{C}}}(i,j,s)}{S_d^{\mathrm{m}}(i,j,s)}. \tag{B12}$$

The second parameter is a normalisation used for when the CHESS-SCAPE value is larger than its local and seasonal mean. This parameter is based on the distance between the mean and the maximum value of CHESS-SCAPE for that location and season,

$$S_d^{\mathrm{C,max}}i,j,s = \max(S_d^{\mathrm{C}}(i,j,k,s,l))|_{k,l}. \tag{B13}$$

The second parameter is thus given by:

$$\mu_{S,2}(i,j,s) = \frac{\overline{S_d^{\mathrm{C}}}(i,j,s) - S_d^{\mathrm{C,max}}(i,j,s)}{S_d^{\mathrm{m}}(i,j,s) - S_d^{\mathrm{C,max}}(i,j,s)}. \tag{B14}$$

We combined this to give the bias-correction as follows:

$$S_d^{\mathrm{BC}}(i,j,k,s,l) = \begin{cases} \dfrac{S_d^{\mathrm{C}}(i,j,k)}{\mu_{S,1}(i,j,s)} & \text{if } S_d^{\mathrm{C}}(i,j,k,s,l) < \overline{S_d^{\mathrm{C}}}(i,j,s) \\ S_d^{\mathrm{C,max}}(i,j,s) + \dfrac{S_d^{\mathrm{C}}(i,j,k,s,l) - S_d^{\mathrm{C,max}}(i,j,s)}{\mu_{S,2}(i.j)} & \text{otherwise.} \end{cases} \tag{B15}$$

This ensured that the seasonal mean of CHESS-SCAPE matched the seasonal mean of CHESS-met. Where the CHESS-SCAPE mean was smaller than the CHESS-met mean this stretched the distribution below the mean and increased the values, while compressing the distribution above the mean to fit in to the range between the CHESS-met mean and maximum. Where the CHESS-SCAPE mean was larger than the CHESS-met mean, this did the opposite and compressed the lower part of the distribution while stretching the upper part.

## Appendix C:  Alternative climate scenarios

### C1   Linearity assumptions

To generate alternative warming scenarios we used a combination of time shifting and pattern scaling (see eg James et al., 2017). Both of these require the assumption that seasonal anomalies in UK variables are linear with respect to global air temperature,

$$\overline{x}(i,j,k,s) = D_x(i,j,s)\overline{T}_G(k), \tag{C1}$$

where $\overline{T}_G(k)$ is annual mean global mean air temperature in year $k$, $D_x(i,j,s)$ is the gradient of the variable $x(i,j,k,s)$ with respect to the global mean air temperature and $\overline{x}(i,j,k,s)$ is the seasonal mean anomaly with respect to the baseline period (1981-2000) of variable $x$ at location $(i,j)$ for season $s$ in year $k$.

For some variables it is more appropriate instead to consider the relative anomaly, particularly where an absolute anomaly could result in unphysical negative values. The relative anomaly is given by

$$\frac{\overline{x}(i,j,k,s)}{\overline{x}_{\text{ref}}(i,j,s)} = D_x^r(i,j,s)\overline{T}_G(k), \tag{C2}$$

where $\overline{x}_{\text{ref}}(i,j,s)$ is the baseline mean of the variable and $D_x^r(i,j,s)$ is the gradient of the relative anomaly with respect to global mean air temperature.

### C2   Time shifting

For each year of each target CMIP5 scenario (RCP2.6, RCP4.5 or RCP6.0), we found all years of the RCP8.5 scenario for which the global annual mean air temperature anomaly was within a threshold $\Delta T = 0.5K$ of the target global annual mean air temperature anomaly, and randomly selected one year for substitution. Once this initial series was defined, we checked each year for repeats within 20 years. If a repeat was found, we discarded the originally selected year performed the random selection again. We iterated this until a timeseries of year substitutions with no repeats within 20 years was found. We then copied the 1km data from each of the RCP8.5 years to the target year to create the time shifted scenario. Note that years were defined to be season-years, running December to January.

### C3   Pattern scaling

We calculated the gradient $D_x(i,j,s)$ or $D_x^r(i,j,s)$ for each variable using linear regression of the RCP8.5 scenario to the linear or relative anomalies, where the UK variables are the CHESS-SCAPE 1 km variables and the global mean air temperature is taken from the corresponding UKCP18 GCM-PPE ensemble member. Then we found the difference between the CMIP5 original RCP8.5 temperature and target RCP temperature. We used this with the seasonal maps of the gradients to calculate new time shifted and pattern scaled time seris.

For air temperature, specific humidity, surface air pressure, daily minimum air temperature and daily maximum air tempera-ture, we calculated the resulting time series of the variable $x'(i,j,k,s,t)$ for location $(i,j)$, year $k$, season $s$ and day of season

$t$ using

$$x'(i,j,k,s,t) = x(i,j,k,s,t) + D_x(i,j,s)\left(\overline{T}'_G(k) - \overline{T}_G(k)\right), \tag{C3}$$

where $\overline{T}_G(k)$ is the global mean air temperature in the time shifted year $k$ and $\overline{T}'_G(k)$ is the global mean air temperature in the target RCP timeseries.

We scaled precipitation, wind speed, downwelling SW radiation and downwelling LW radiation using relative anomalies:

$$x'(i,j,k,s,t) = x(i,j,k,s,t)\left(1 + D_x^r(i,j,s)\left(\overline{T}'_G(k) - \overline{T}_G(k)\right)\right). \tag{C4}$$

*Author contributions.* ELR developed and applied the downscaling code, and the time shifting and pattern scaling code. CHG developed and applied the bias correction code. SVS carried out checking of the data files. All authors developed the methods and wrote the manuscript.

*Competing interests.* The authors declare no competing interests.

*Acknowledgements.* This research was carried out under national capability funding as part of the Spatially explicit projections of environmental drivers and impacts (SPEED) work package of the Natural Environment Research Council UK-SCAPE programme, award number NE/R016429/1. Additional support provided by the Natural Environment Research Council award number NE/S017380/1 as part of the Hydro-JULES programme.

The authors would like to thank Fai Fung and Jason Lowe of the UK Met Office for valuable discussions and advice while developing this
dataset.

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

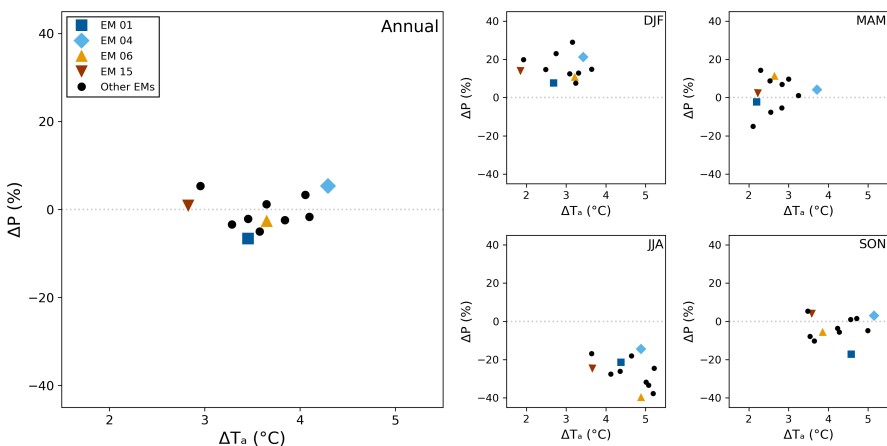

**Figure 1.** Change in precipitation and air temperature between 1980 – 2000 and 2060 – 2080, given by the regional mean of the UKCP18 RCM-PPE 12 km output. The main panel shows the change in the annual mean, the smaller panels show the change in the seasonal mean for each season (DJF, MAM, SON, JJA). Highlighted ensemble members 01, 04, 15 and 06 were selected to be the basis of the CHESS-SCAPE 1 km output dataset.

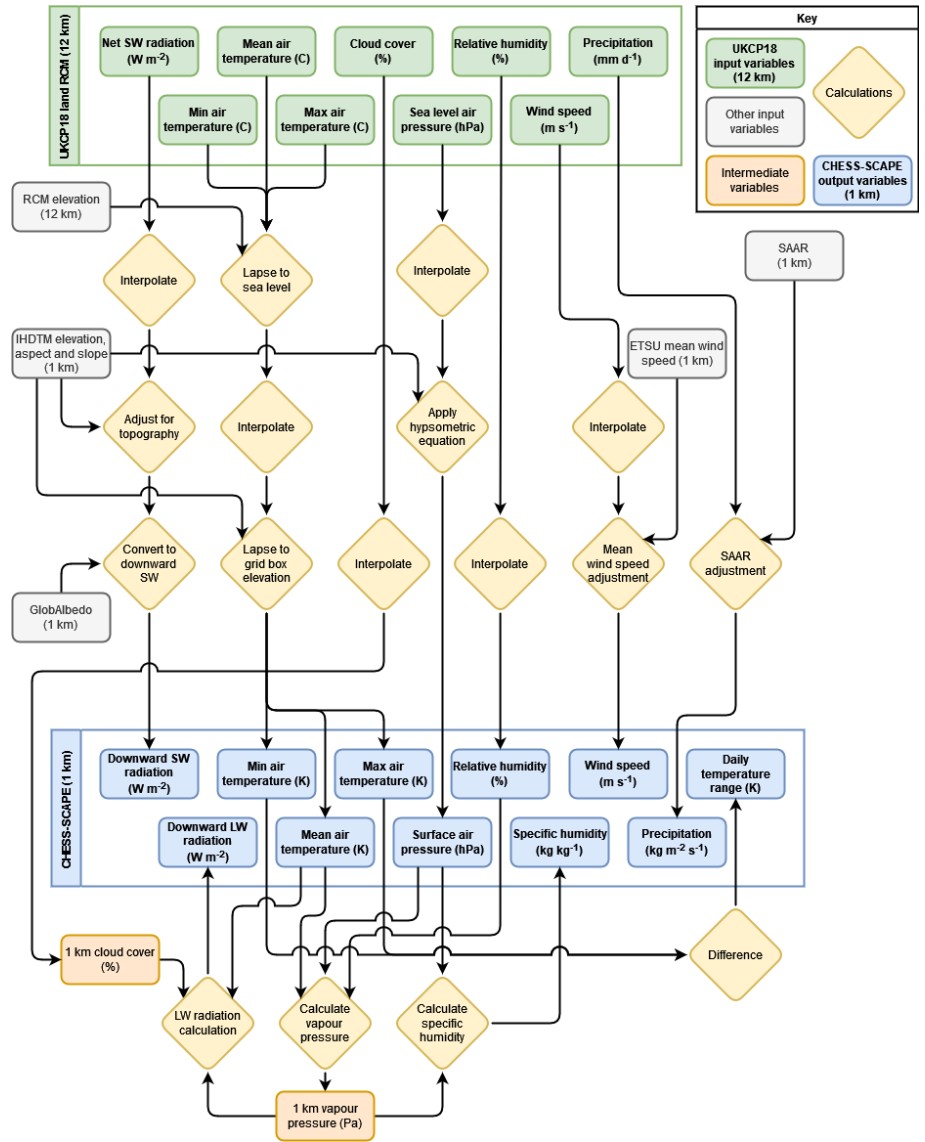

**Figure 2.** Flow diagram showing the downscaling procedure.

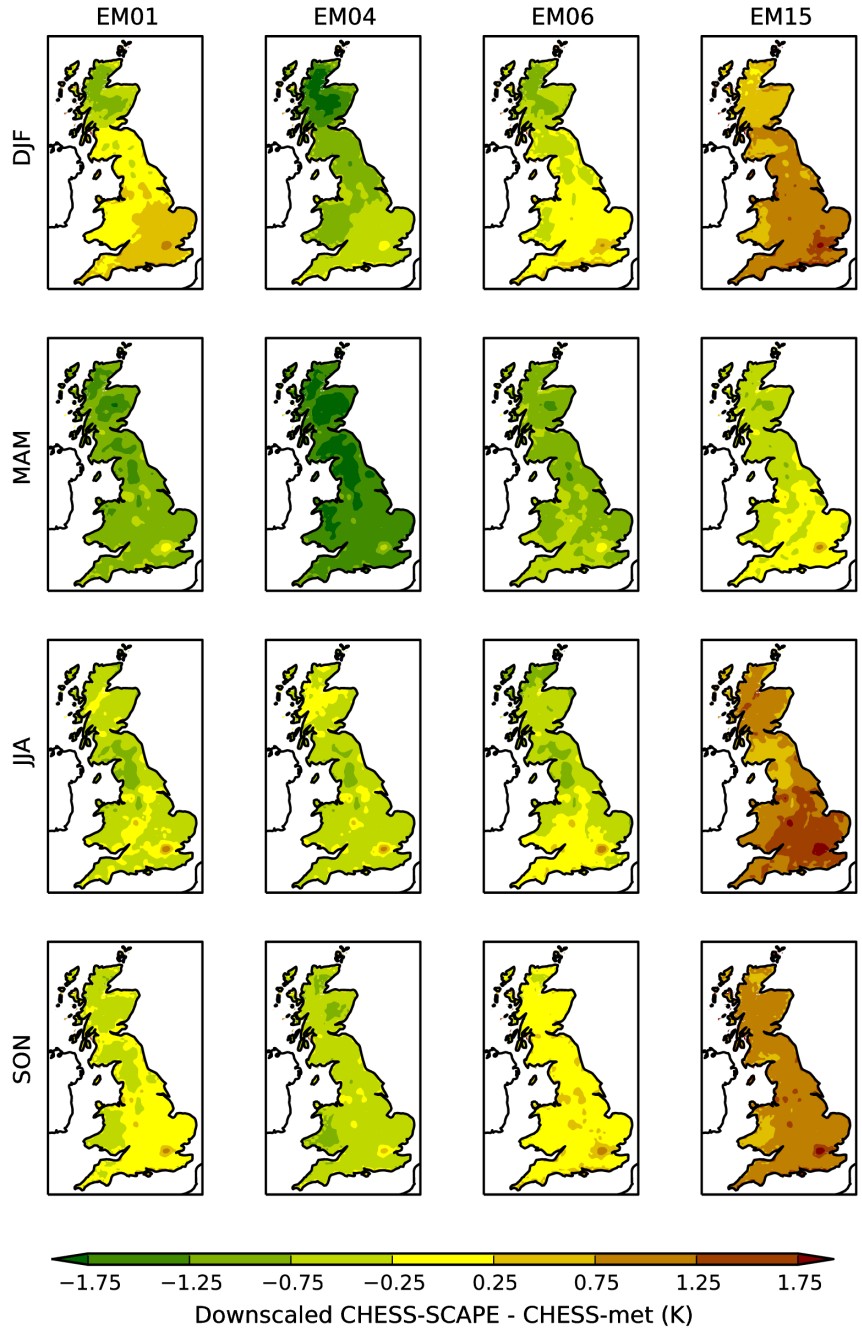

**Figure 3.** Difference in air temperature between the downscaled CHESS-SCAPE data and the historical CHESS-met data, $\mu_T$, for each season and each ensemble member.

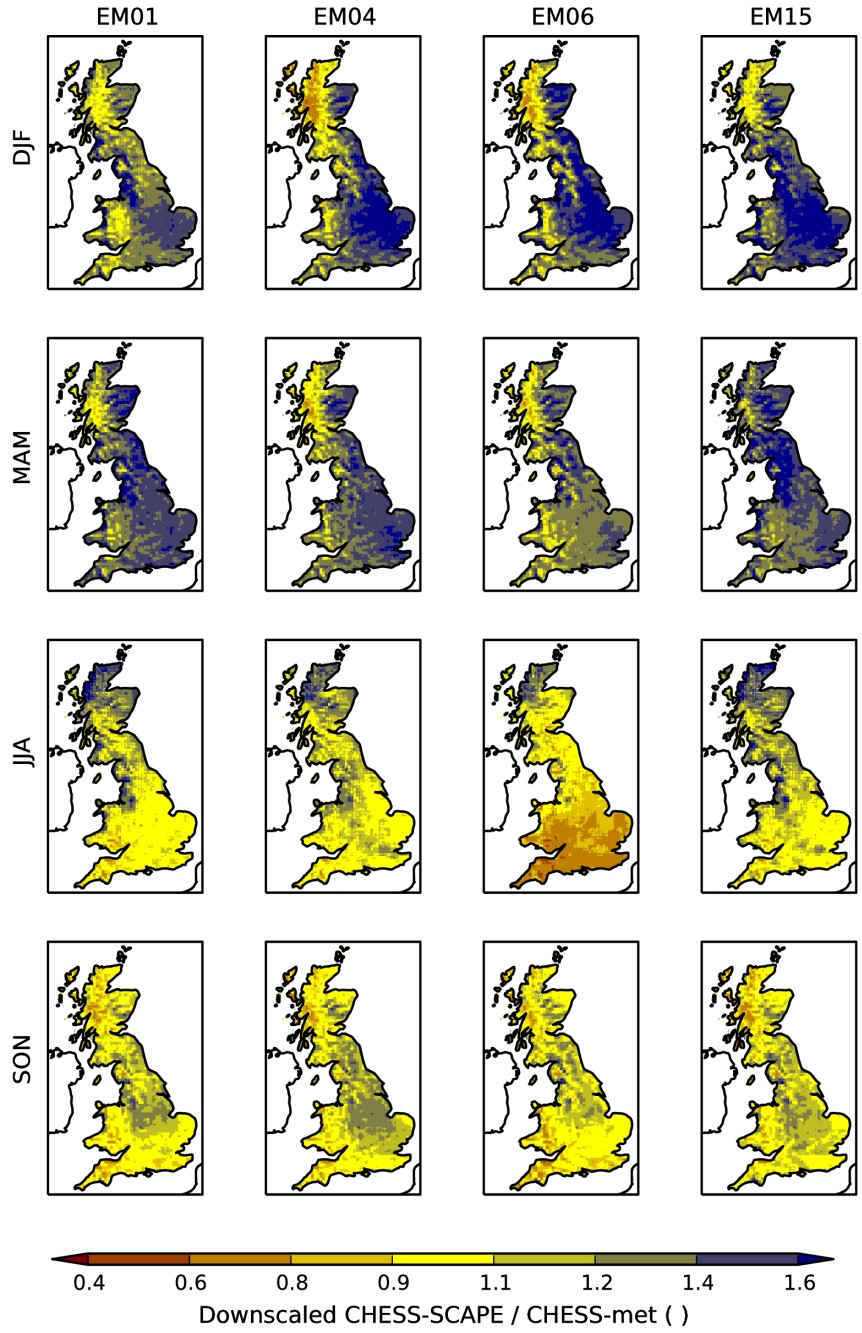

**Figure 4.** Ratio of precipitation in the downscaled CHESS-SCAPE data and the historical CHESS-met data, $\mu_P$, for each season and each ensemble member.

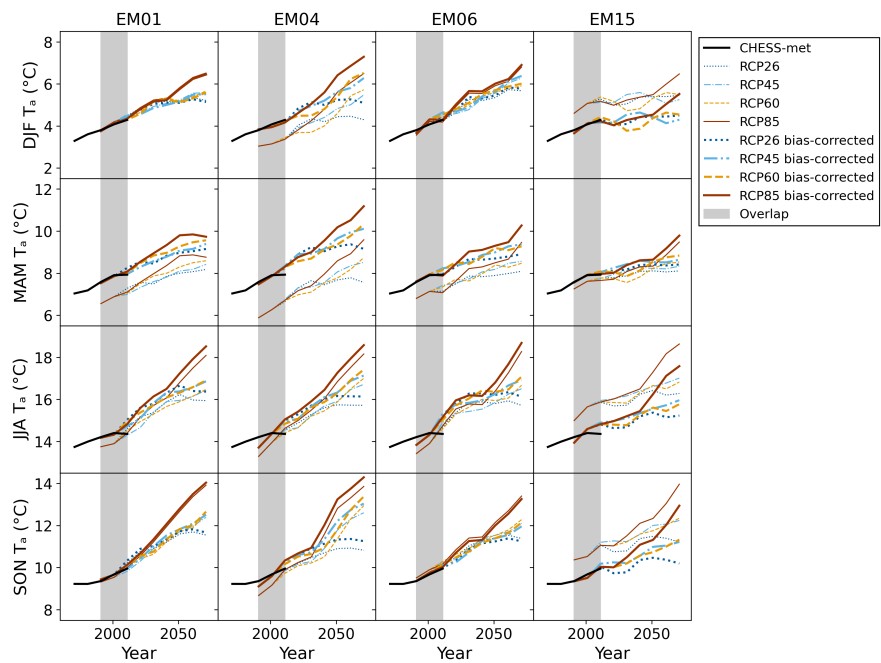

**Figure 5.** Twenty-year seasonal mean air temperature over GB for the downscaled-only CHESS-SCAPE data (thin coloured lines), the downscaled and bias-corrected CHESS-SCAPE data (thick coloured lines), and CHESS-met (black). The different RCPs are RCP2.6 (dark blue dotted), RCP4.5 (light blue dash-dotted), RCP6.0 (yellow dashed) and RCP8.5 (brown). The grey region shows the period of overlap 1961–2015.

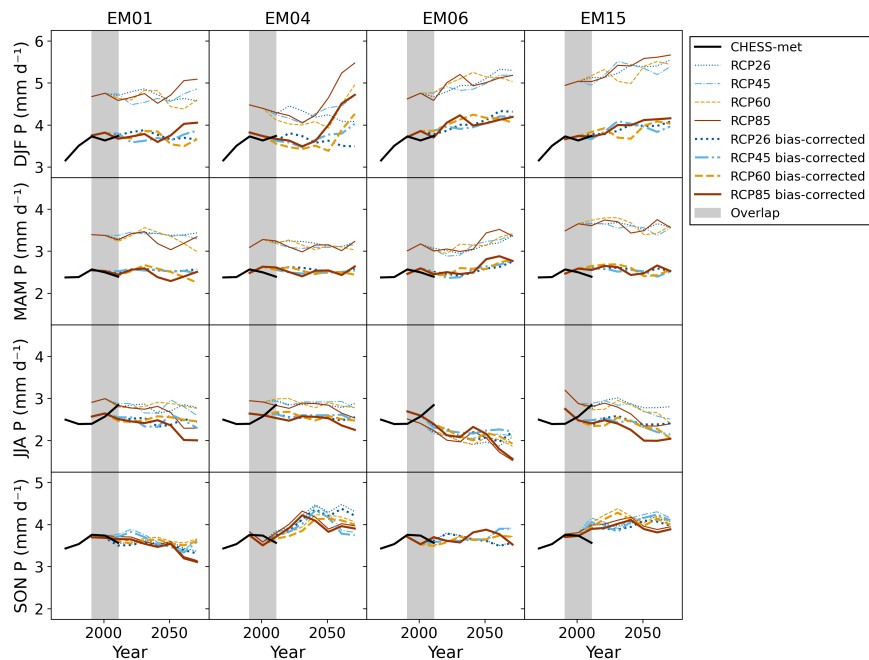

**Figure 6.** Twenty-year seasonal mean precipitation over GB for the downscaled-only CHESS-SCAPE data (thin coloured lines), the downscaled and bias-corrected CHESS-SCAPE data (thick coloured lines), and CHESS-met (black). The different RCPs are RCP2.6 (dark blue dotted), RCP4.5 (light blue dash-dotted), RCP6.0 (yellow dashed) and RCP8.5 (brown). The grey region shows the period of overlap 1961–2015.

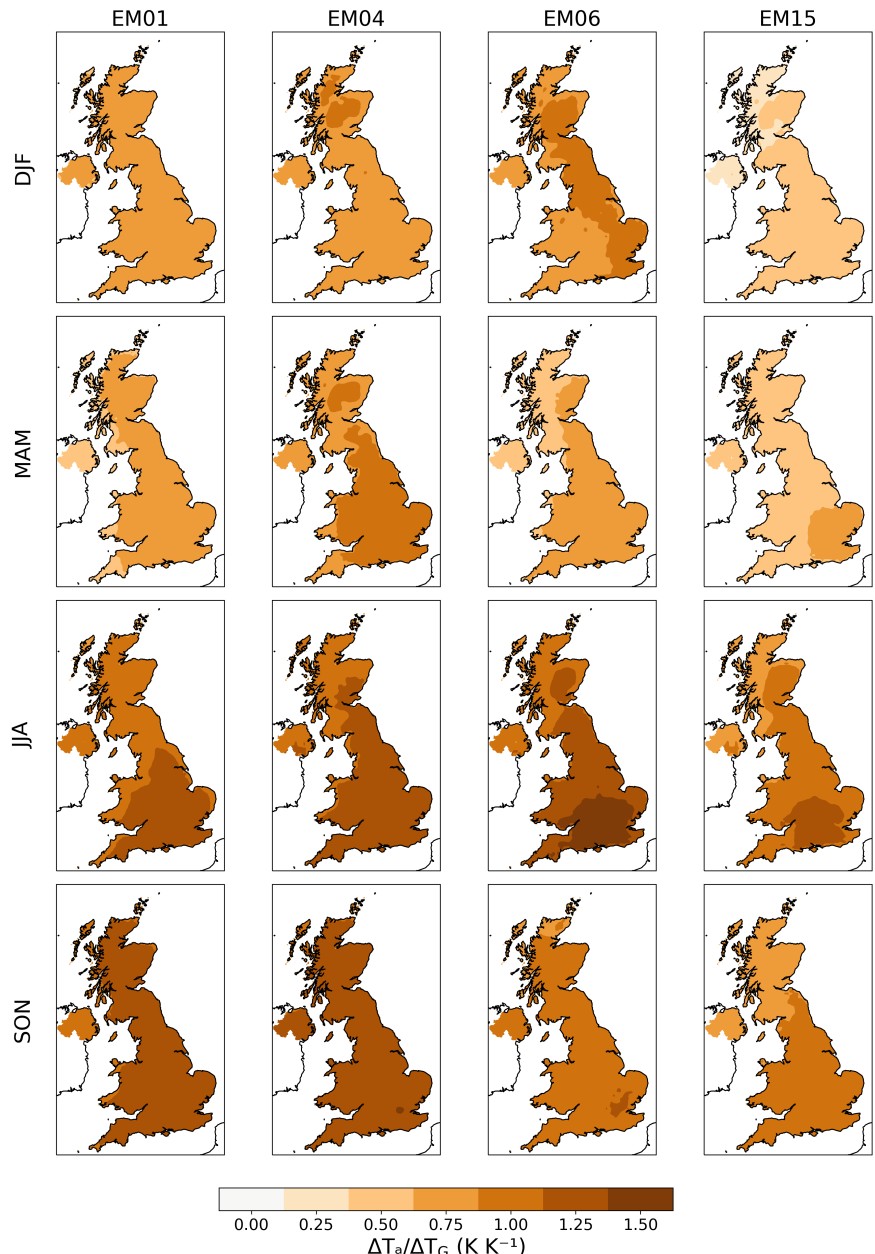

**Figure 7.** Gradient of linear fit of seasonal mean air temperature to global annual mean air temperature for each 1 km pixel for each downscaled-only ensemble member

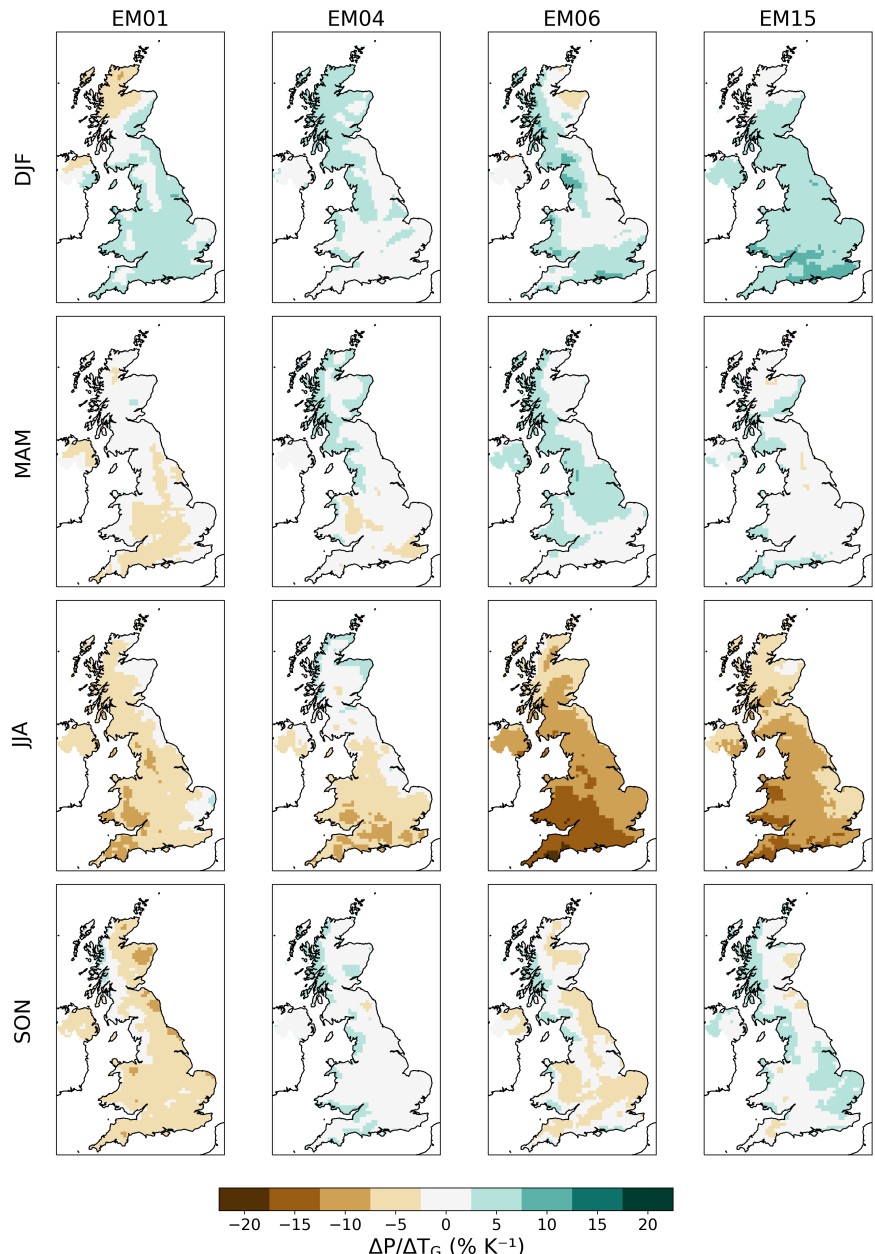

**Figure 8.** Gradient of linear fit of seasonal mean precipitation to global annual mean air temperature for each 1 km pixel for each downscaled-only ensemble member

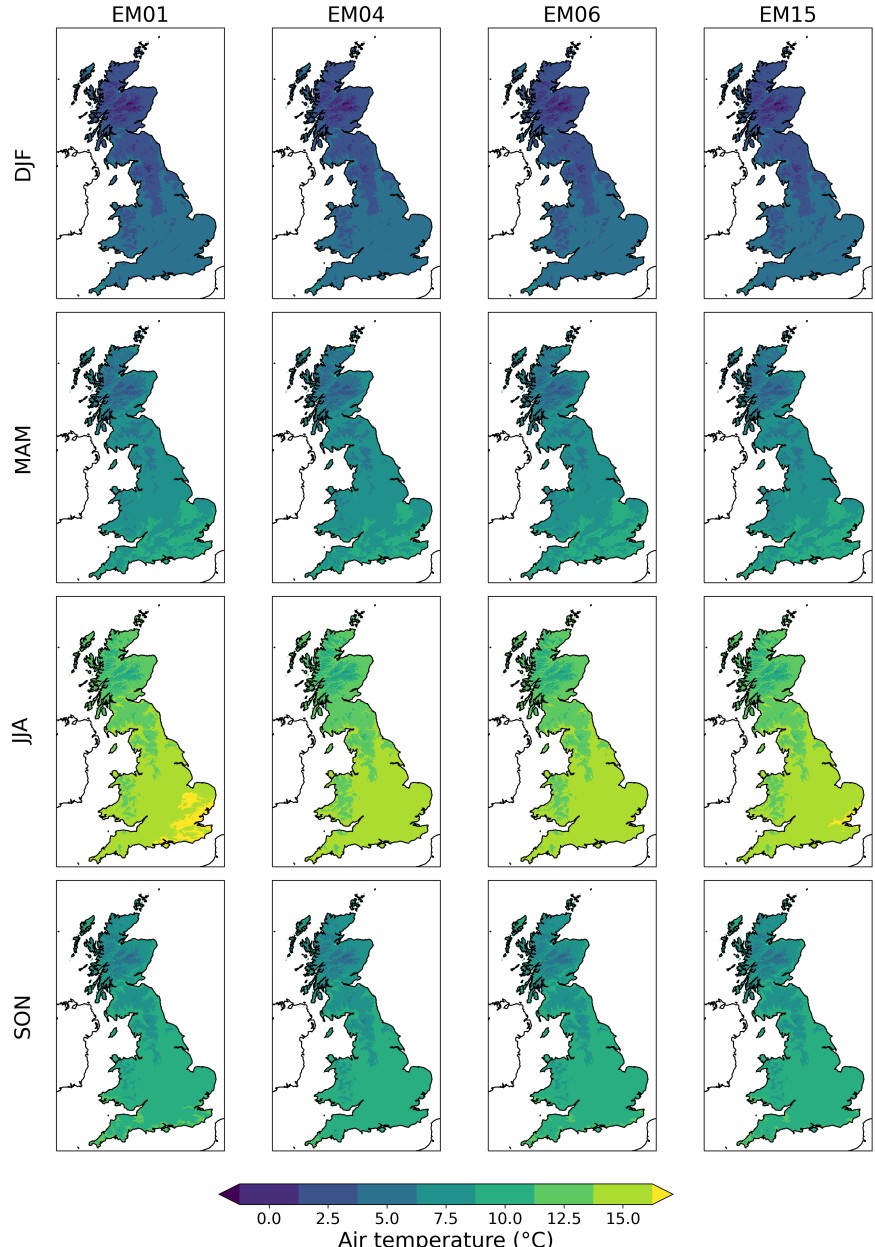

**Figure 9.** Seasonal mean bias-corrected air temperature 1980 – 2000.

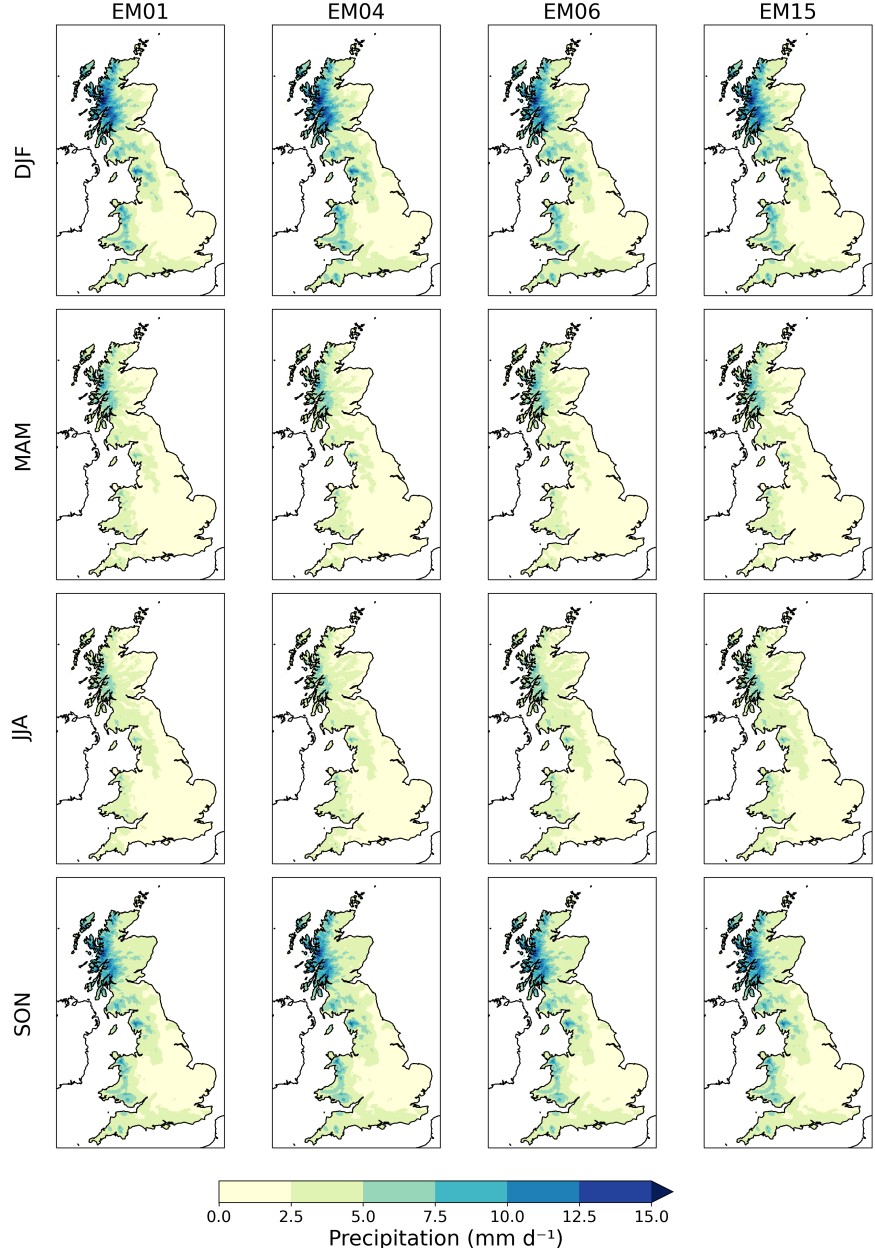

**Figure 10.** Seasonal mean bias-corrected precipitation 1980 – 2000.

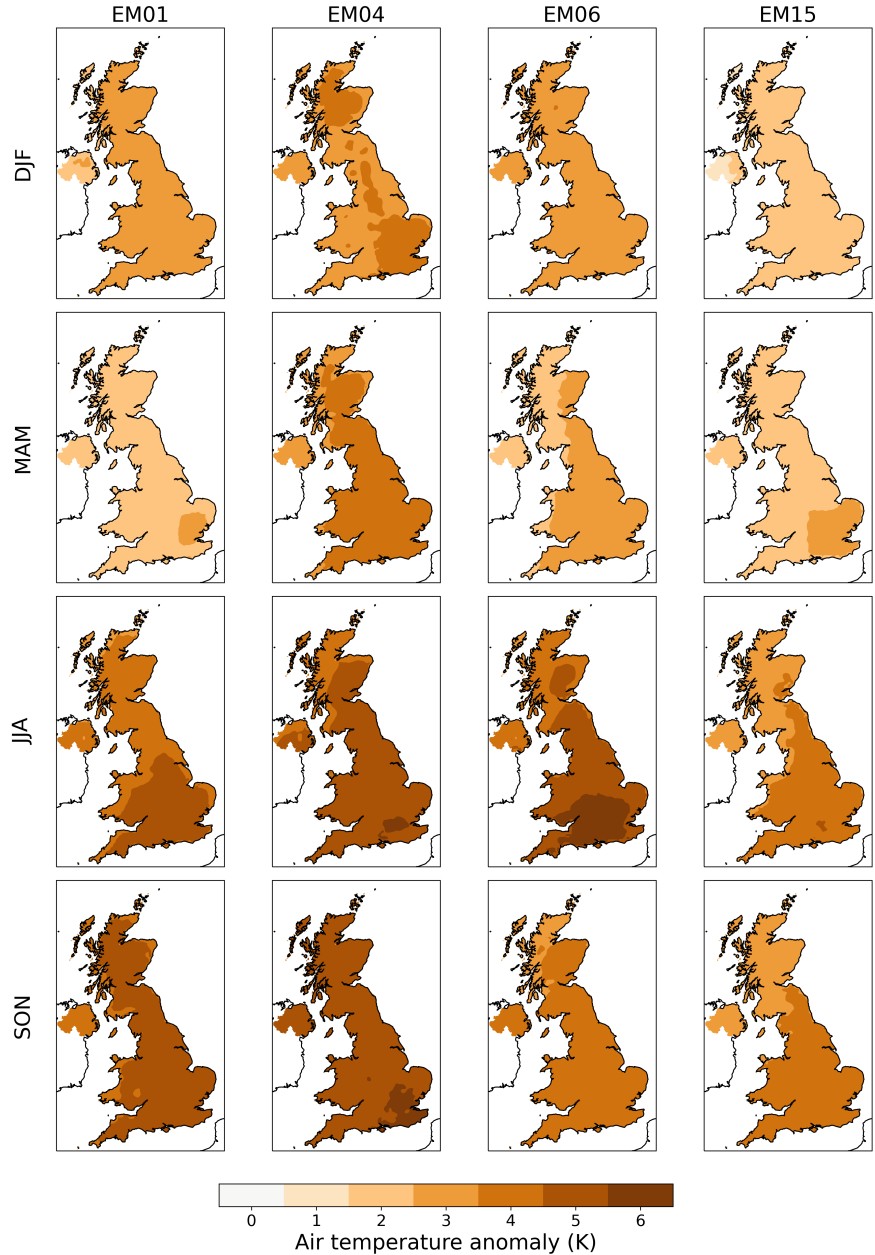

**Figure 11.** Seasonal air temperature anomalies 2060 – 2080 with respect to the baseline period 1980 – 2000 for downscaled-only RCP8.5.

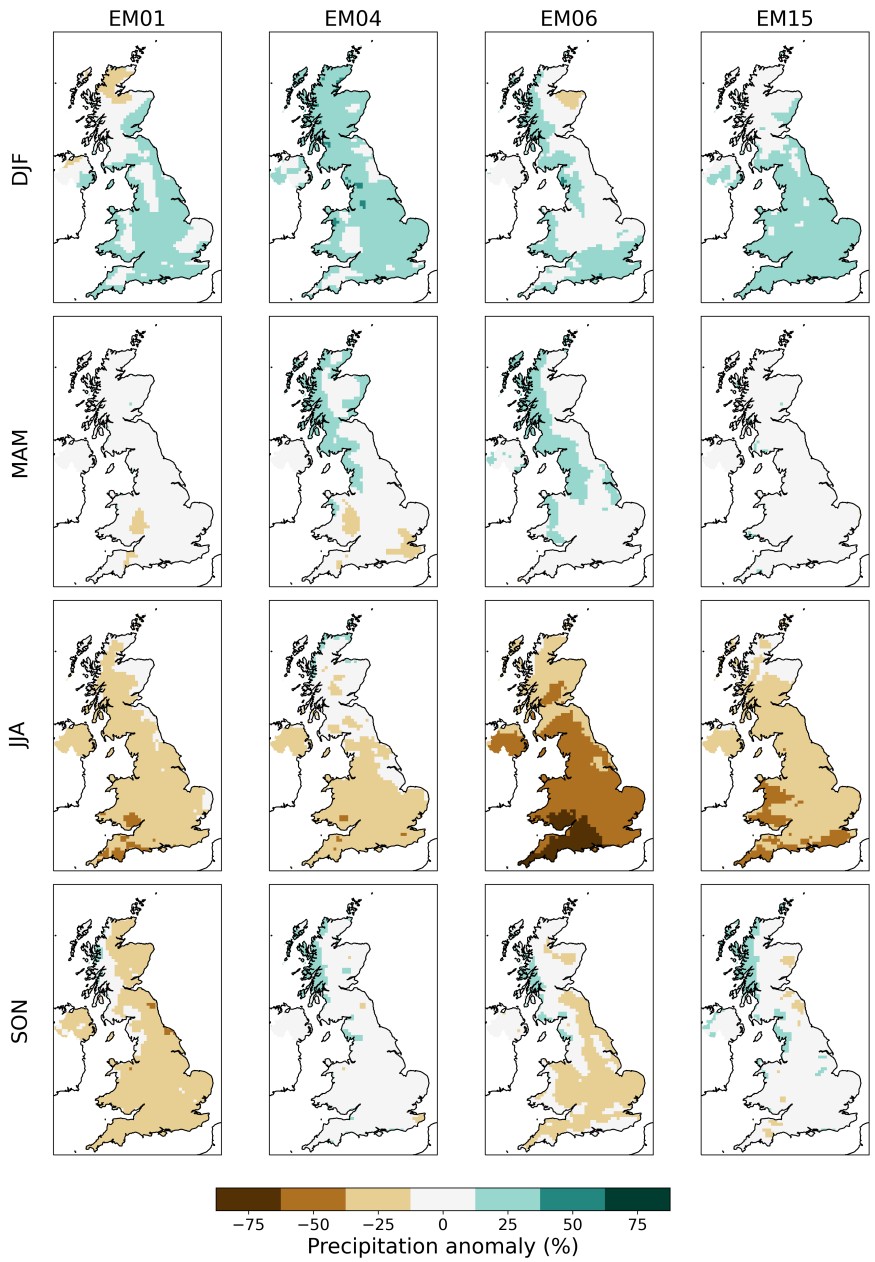

**Figure 12.** Seasonal precipitation anomalies 2060 – 2080 with respect to the baseline period 1980 – 2000 for downscaled-only RCP8.5.

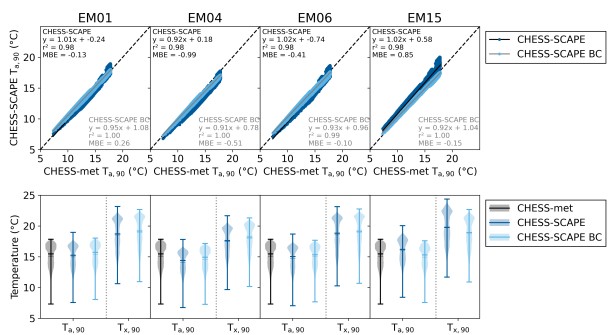

**Figure 13.** 90th percentile values of daily mean air temperature, $T_{a,90}^{\mathrm{C}}$ for the period 1980–2000. The top row shows the CHESS-SCAPE values $T_{a,90}^{\mathrm{C}}$ for each ensemble member plotted against the CHESS-met values $T_{a,90}^{\mathrm{m}}$ for each corresponding 1 km grid box for the downscaled-only data (dark blue) and the bias-corrected data (light blue). The bottom row shows violin plots of the $T_{a,90}$ values for CHESS-met (black), CHESS-SCAPE downscaled-only (dark blue) and CHESS-SCAPE bias-corrected (light blue). The thick horizontal lines show the median value and the extremes, the thin horizontal lines show the mean.

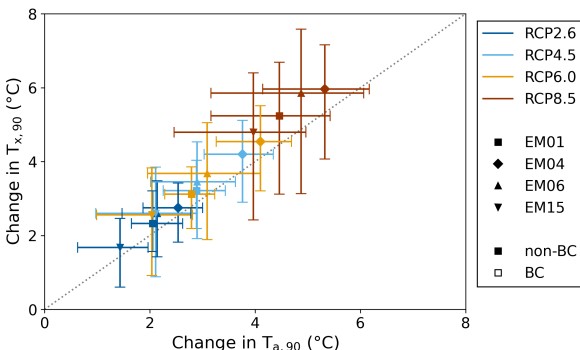

**Figure 14.** Difference between the 1980–2000 and 2060–2080 90th percentile values of downscaled-only daily maximum air temperature plotted against the difference between the 1980–2000 and 2060–2080 90th percentile values of downscaled-only daily mean air temperature, for RCP2.6 (dark blue), RCP4.5 (light blue), RCP6.0 (yellow) and RCP8.5 (brown) for each ensemble member 01 (square), 04 (diamond), 06 (upward triangle) and 15 (downward triangle). Symbols indicate the mean, error bars show the range over the available pixels. The dashed grey line shows the 1:1 correspondence.

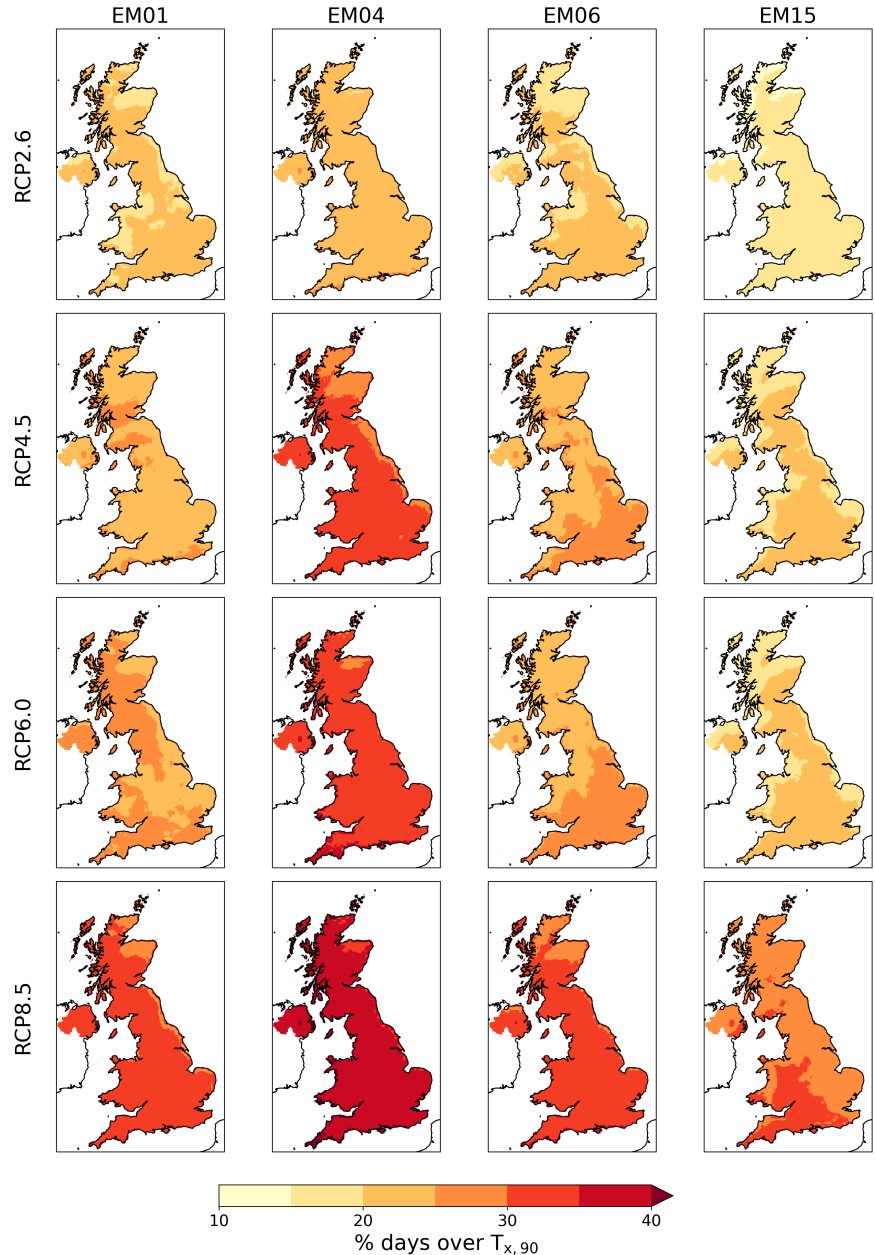

**Figure 15.** Percentage of days between 2060–2080 for which the daily maximum air temperature exceeds the 90th percentile of daily maximum air temperature for 1980–2080 in the downscaled-only data, for each ensemble member and each RCP.

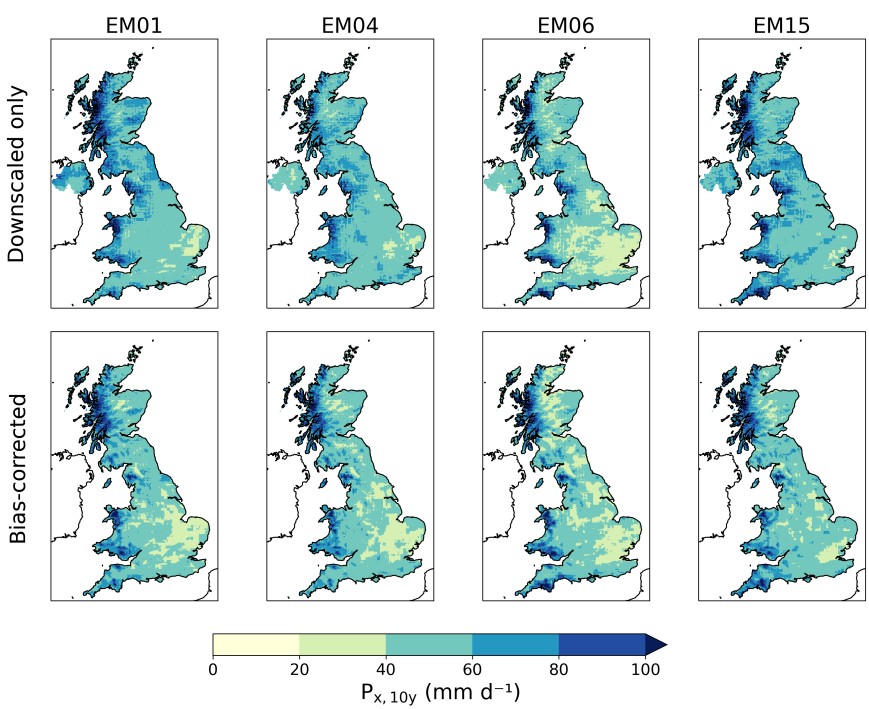

**Figure 16.** 10-year return levels of annual maximum precipitation for the period 1980–2010 for each bias-corrected ensemble member and RCP.

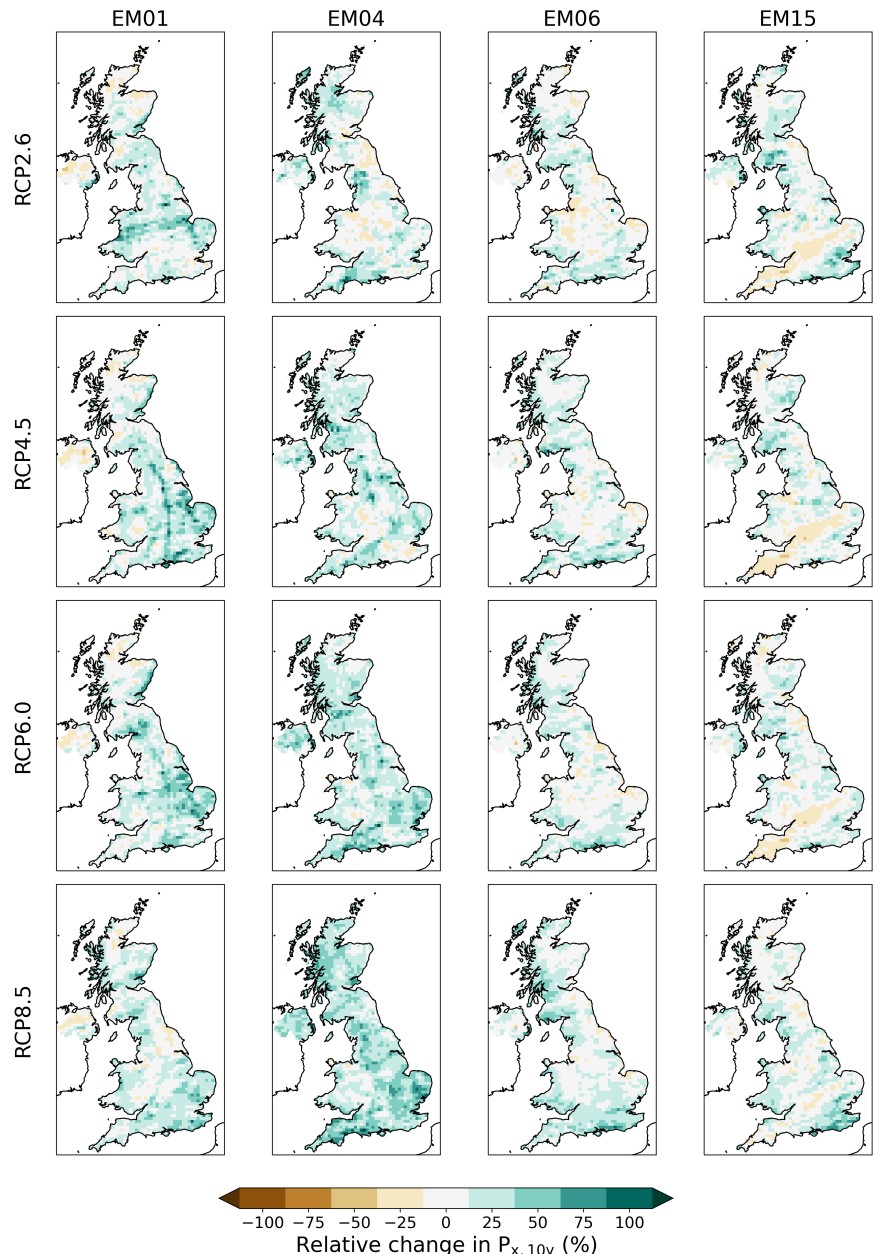

**Figure 17.** Percentage change in 10-year return levels of annual maximum precipitation between 1980–2010 and 2050–2080 for each downscaled-only ensemble member and RCP.

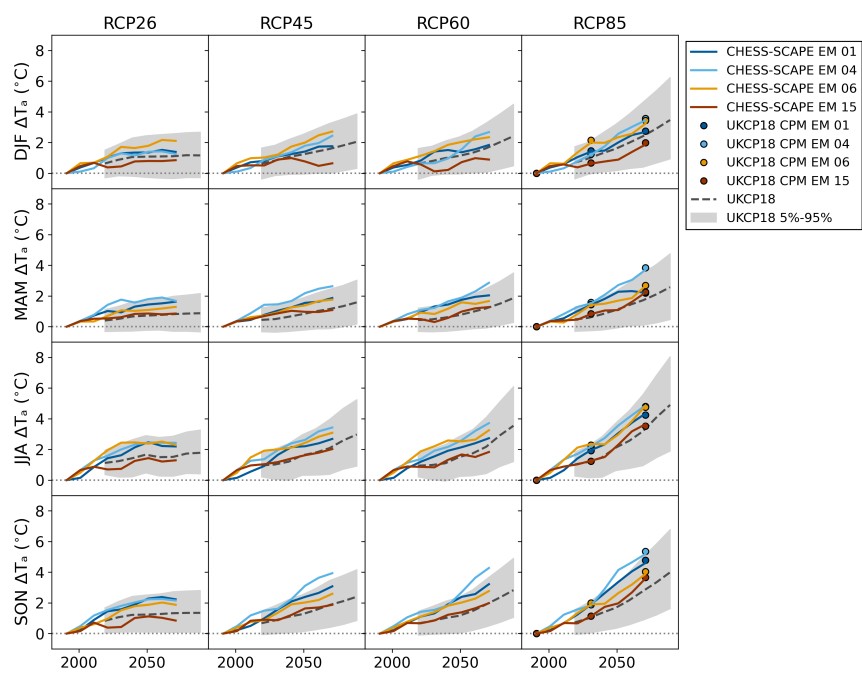

**Figure 18.** Twenty-year seasonal mean air temperature anomalies for each ensemble member and RCP. The coloured lines show CHESS-SCAPE bias-corrected variables RCP2.6 (dark blue), RCP4.5 (light blue), RCP6.0 (yellow), RCP8.5 (brown). The grey lines show the median of the UKCP18 probabilistic projections, while the light grey regions show the 5%-95% interval. The dots show the mean of the UKCP18 CPM-PPE projections at 1980-2000, 2020-2040 and 2060-2080.

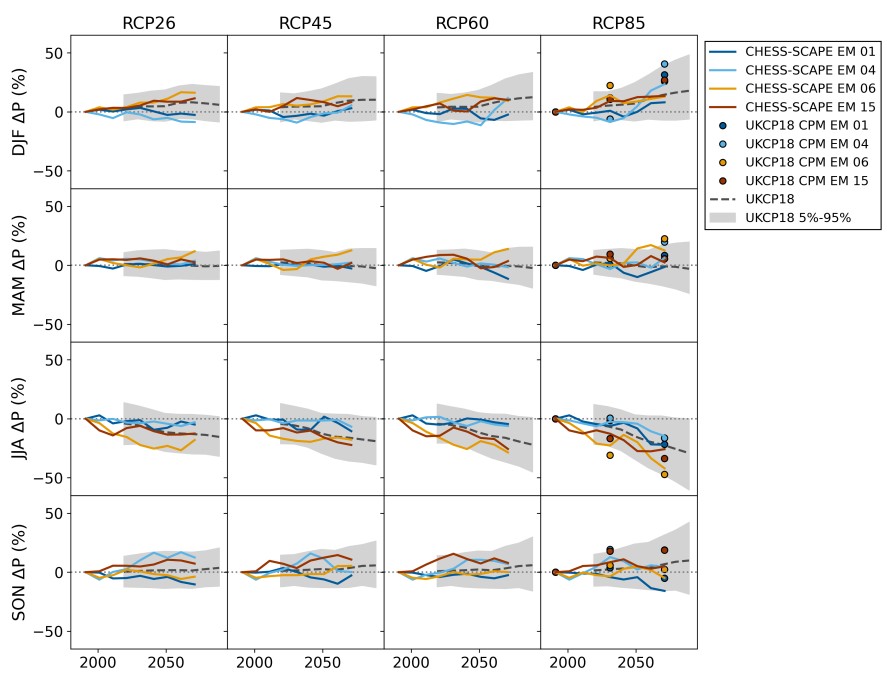

**Figure 19.** Twenty-year seasonal mean precipitation anomalies for each ensemble member and RCP. The coloured lines show CHESS-SCAPE bias-corrected variables RCP2.6 (dark blue), RCP4.5 (light blue), RCP6.0 (yellow), RCP8.5 (brown). The grey lines show the median of the UKCP18 probabilistic projections, while the light grey regions show the 5%-95% interval. The dots show the mean of the UKCP18 CPM-PPE projections at 1980-2000, 2020-2040 and 2060-2080.

| Long name | Symbol | Units | Short name | CF standard name |
|---|---|---|---|---|
| Cloud area fraction | $C_f^{\mathrm{U}}$ | % | clt | cloud_area_fraction |
| Relative humidity[*] | $R^{\mathrm{U}}$ | % | hurs | relative_humidity |
| Specific humidity[*] | $q_a^{\mathrm{U}}$ | 1 | huss | specific_humidity |
| Precipitation rate | $P^{\mathrm{U}}$ | mm/day | pr | lwe_precipitation_rate |
| Sea level pressure | $p_*^{\mathrm{U}}$ | hPa | psl | air_pressure_at_sea_level |
| Net Surface SW flux | $S_n^{\mathrm{U}}$ | $\mathrm{W\,m^{-2}}$ | rss | surface_net_downward_shortwave_flux |
| Wind speed[†] | $u^{\mathrm{U}}$ | $\mathrm{m\,s^{-1}}$ | sfcWind | wind_speed |
| Mean air temperature[*] | $T_a^{\mathrm{U}}$ | °C | tas | air_temperature |
| Maximum air temperature[*] | $T_n^{\mathrm{U}}$ | °C | tasmax | air_temperature |
| Minimum air temperature[*] | $T_x^{\mathrm{U}}$ | °C | tasmin | air_temperature |

[*] 1.5 m above surface.

[†] 10 m above surface.

**Table 1.** Variables in the UKCP18 RCM-PPE 12 km data.

| Long name | Symbol | Units | Short name | CF standard name | Bias corrected |
|---|---|---|---|---|---|
| Daily near-surface air temperature range* | $\Delta_T^C$ | K | dtr | air_temperature | N |
| Near-surface relative humidity | $R^C$ | % | hurs | relative_humidity | Y |
| Near-surface specific humidity | $q_a^C$ | 1 | huss | specific_humidity | Y |
| Precipitation flux | $P^C$ | $\text{kg m}^{-2}\,\text{s}^{-1}$ | pr | precipitation_flux | Y |
| Surface air pressure | $p_*^C$ | Pa | psurf | surface_air_pressure | N |
| Surface downwelling longwave radiation | $L_d^C$ | $\text{W m}^{-2}$ | rlds | surface_downwelling_longwave_flux_in_air | Y |
| Surface downwelling shortwave radiation | $L_n^C$ | $\text{W m}^{-2}$ | rsds | surface_downwelling_shortwave_flux_in_air | Y |
| Wind speed† | $u^C$ | $\text{m s}^{-1}$ | sfcWind | wind_speed | Y |
| Near-surface air temperature* | $T_a^C$ | K | tas | air_temperature | Y |
| Near-surface daily maximum air temperature* | $T_n^C$ | K | tasmax | air_temperature | Y |
| Near-surface daily minimum air temperature* | $T_x^C$ | K | tasmin | air_temperature | Y |

* 1.5 m above surface.
† 10 m above surface.

**Table 2.** Variables in the CHESS-SCAPE 1 km resolution dataset.

| Long name | Units | Short name | CF standard name |
|---|---|---|---|
| Daily air temperature range | K | dtr | air_temperature |
| Specific humidity* | 1 | huss | specific_humidity |
| Precipitation flux | kg m$^{-2}$ s$^{-1}$ | precip | precipitation_flux |
| Surface air pressure | Pa | psurf | surface_air_pressure |
| Surface downwelling longwave radiation | W m$^{-2}$ | rlds | surface_downwelling_<br>longwave_flux_in_air |
| Surface downwelling SW radiation | W m$^{-2}$ | rsds | surface_downwelling_<br>shortwave_flux_in_air |
| Near-surface wind speed[†] | m s$^{-1}$ | sfcWind | wind_speed |
| Near-surface air temperature* | K | tas | air_temperature |

[*] 1.2 m above surface.

[†] 10 m above surface.

**Table 3.** Variables in the CHESS-met data.

**Table 4.** Climate change in ensemble members between the baseline period (1980 – 2000) and the end of the climate projections (2060 – 2080).

| Ensemble member | Air temperature (K) | | | | | Precipitation (%) | | | | |
|---|---|---|---|---|---|---|---|---|---|---|
| | Annual | DJF | MAM | JJA | SON | Annual | DJF | MAM | JJA | SON |
| 01* | 3.5 | 2.7 | 2.2 | 4.4 | 4.6 | -7 | 8 | -2 | -21 | -17 |
| 04* | 4.3 | 3.4 | 3.7 | 4.9 | 5.1 | 5 | 21 | 4 | -14 | 3 |
| 05 | 3.8 | 3.2 | 3.2 | 4.6 | 4.2 | -2 | 8 | 1 | -18 | -4 |
| 06* | 3.6 | 3.2 | 2.6 | 4.9 | 3.9 | -3 | 11 | 11 | -40 | -6 |
| 07 | 3.0 | 1.9 | 2.3 | 4.1 | 3.5 | 5 | 20 | 14 | -27 | 5 |
| 08 | 3.6 | 2.5 | 2.5 | 5.0 | 4.6 | 1 | 15 | 9 | -32 | 1 |
| 09 | 4.1 | 3.2 | 3.0 | 5.1 | 5.0 | 3 | 29 | 10 | -33 | -5 |
| 10 | 3.5 | 3.1 | 2.8 | 4.4 | 3.5 | -2 | 13 | 7 | -26 | -8 |
| 11 | 4.1 | 3.6 | 2.8 | 5.2 | 4.7 | -2 | 15 | -5 | -25 | 2 |
| 12 | 3.3 | 3.3 | 2.5 | 3.6 | 3.6 | -3 | 13 | -8 | -17 | -10 |
| 13 | 3.6 | 2.7 | 2.1 | 5.2 | 4.3 | -5 | 23 | -15 | -38 | -6 |
| 15* | 2.8 | 1.8 | 2.2 | 3.7 | 3.6 | 1 | 14 | 2 | -25 | 4 |

* Ensemble members used for CHESS-SCAPE

**Table 5.** CMIP5 models selected as proxies for intermediate RCPs

| UKCP18 ensemble member | Institution | Model | Ensemble member | KGE |
|---|---|---|---|---|
| 01 | MOHC | HadGEM2-ES | r4i1p1 | 0.87 |
| 04 | MOHC | HadGEM2-ES | r2i1p1 | 0.78 |
| 06 | MOHC | HadGEM2-ES | r1i1p1 | 0.85 |
| 15 | IPSL | IPSL-CM5A-MR | r1i1p1 | 0.93 |

**Table 6.** 2060 – 2080 air temperature anomaly with respect to the baseline period 1980 – 2000 for each CHESS-SCAPE ensemble member and scenario for the downscaled-only data (mean over the whole UK) and the downscaled and bias-corrected data (mean over GB only)

| Ensemble member | RCP | Air temperature anomaly (K) | | | | | Bias-corrected Air temperature anomaly (K) | | | | |
|---|---|---|---|---|---|---|---|---|---|---|---|
| | | Annual | DJF | MAM | JJA | SON | Annual | DJF | MAM | JJA | SON |
| 01 | 2.6 | 1.8 | 1.4 | 1.6 | 2.2 | 2.2 | 1.9 | 1.4 | 1.6 | 2.2 | 2.2 |
| 01 | 4.5 | 2.3 | 1.7 | 1.9 | 2.7 | 3.1 | 2.3 | 1.8 | 1.9 | 2.7 | 3.1 |
| 01 | 6.0 | 2.4 | 1.8 | 2.0 | 2.7 | 3.2 | 2.5 | 1.8 | 2.0 | 2.7 | 3.2 |
| 01 | 8.5 | 3.4 | 2.7 | 2.2 | 4.3 | 4.6 | 3.5 | 2.7 | 2.2 | 4.3 | 4.6 |
| 04 | 2.6 | 1.9 | 1.2 | 1.7 | 2.4 | 2.1 | 1.9 | 1.3 | 1.7 | 2.4 | 2.2 |
| 04 | 4.5 | 3.1 | 2.4 | 2.6 | 3.4 | 3.9 | 3.1 | 2.4 | 2.6 | 3.4 | 3.9 |
| 04 | 6.0 | 3.4 | 2.7 | 2.8 | 3.7 | 4.3 | 3.4 | 2.7 | 2.9 | 3.7 | 4.3 |
| 04 | 8.5 | 4.3 | 3.4 | 3.7 | 4.8 | 5.2 | 4.3 | 3.4 | 3.7 | 4.9 | 5.2 |
| 06 | 2.6 | 1.9 | 2.1 | 1.3 | 2.3 | 1.9 | 1.9 | 2.1 | 1.3 | 2.3 | 1.9 |
| 06 | 4.5 | 2.5 | 2.7 | 1.7 | 3.1 | 2.6 | 2.5 | 2.7 | 1.8 | 3.1 | 2.6 |
| 06 | 6.0 | 2.5 | 2.3 | 1.6 | 3.2 | 2.8 | 2.5 | 2.3 | 1.7 | 3.2 | 2.8 |
| 06 | 8.5 | 3.6 | 3.2 | 2.6 | 4.8 | 3.9 | 3.7 | 3.2 | 2.7 | 4.9 | 3.9 |
| 15 | 2.6 | 0.9 | 0.8 | 0.8 | 1.3 | 0.8 | 1.0 | 0.8 | 0.9 | 1.3 | 0.8 |
| 15 | 4.5 | 1.4 | 0.6 | 1.1 | 2.0 | 1.9 | 1.4 | 0.6 | 1.1 | 2.0 | 1.9 |
| 15 | 6.0 | 1.5 | 0.9 | 1.3 | 1.8 | 2.0 | 1.5 | 0.9 | 1.3 | 1.8 | 2.0 |
| 15 | 8.5 | 2.8 | 1.8 | 2.2 | 3.6 | 3.6 | 2.8 | 1.9 | 2.2 | 3.6 | 3.6 |

**Table 7.** Relative 2060 – 2080 precipitation anomaly with respect to the baseline period 1980 – 2000 for each CHESS-SCAPE ensemble member and scenario for the downscaled-only data (mean over the whole UK) and the downscaled and bias-corrected data (mean over GB only)

| Ensemble member | RCP | Precipitation anomaly (%) | | | | | Bias-corrected Precipitation anomaly (%) | | | | |
|---|---|---|---|---|---|---|---|---|---|---|---|
| | | Annual | DJF | MAM | JJA | SON | Annual | DJF | MAM | JJA | SON |
| 01 | 2.6 | -4 | -3 | 1 | -4 | -10 | -5 | -3 | 1 | -5 | -10 |
| 01 | 4.5 | -3 | 3 | -3 | -11 | -3 | -3 | 3 | -3 | -11 | -3 |
| 01 | 6.0 | -5 | -2 | -12 | -5 | -3 | -5 | -2 | -11 | -4 | -3 |
| 01 | 8.5 | -6 | 8 | -2 | -21 | -16 | -7 | 8 | -1 | -22 | -16 |
| 04 | 2.6 | 1 | -9 | 4 | -2 | 12 | 1 | -9 | 4 | -3 | 12 |
| 04 | 4.5 | 1 | 5 | 2 | -6 | 0 | 1 | 5 | 2 | -7 | 0 |
| 04 | 6.0 | 3 | 11 | -2 | -5 | 6 | 4 | 12 | -2 | -6 | 7 |
| 04 | 8.5 | 6 | 22 | 4 | -14 | 4 | 7 | 23 | 6 | -15 | 4 |
| 06 | 2.6 | 3 | 14 | 12 | -17 | -4 | 2 | 16 | 12 | -18 | -4 |
| 06 | 4.5 | 5 | 12 | 13 | -16 | 5 | 4 | 13 | 13 | -18 | 5 |
| 06 | 6.0 | 1 | 8 | 14 | -26 | 0 | -1 | 9 | 14 | -29 | 0 |
| 06 | 8.5 | -2 | 11 | 12 | -39 | -5 | -4 | 13 | 13 | -42 | -5 |
| 15 | 2.6 | 4 | 12 | 3 | -12 | 8 | 3 | 11 | 3 | -13 | 7 |
| 15 | 4.5 | 1 | 9 | 2 | -21 | 10 | 1 | 8 | 2 | -22 | 11 |
| 15 | 6.0 | 1 | 10 | 3 | -24 | 8 | 0 | 10 | 4 | -26 | 8 |
| 15 | 8.5 | 1 | 14 | 3 | -25 | 5 | 0 | 14 | 2 | -26 | 5 |