# Peer review of "CHESS-SCAPE: High resolution future projections of multiple climate scenarios for the United Kingdom derived from downscaled UKCP18 regional climate model output"

_Earth System Science Data, 2022_

## Author Response (AR1)

We thank the reviewers for their comments and address them in detail below. We have prepared an updated manuscript, with amendments to the text and an added analysis of extreme values, which we think will improve readers' understanding of the dataset. This includes adding five figures (Figs 13-17), and moving the old Figs 13-14 to become Figs 18-19. We have made some other minor amendments to the text to aid readability, which are all noted in the track-changes version of the manuscript.

Note that additions to the references have not been highlighted in the track-changes version, as latexdiff cannot track changes in bibtex bibliographies.

**Response to RC1:**

*L228 "We based the choice on the mean change in UK near-surface air temperature and precipitation between the baseline period 1980 – 2000 and the final twenty years of the230 future projections 2060 – 2080 (Table 4)". Model selection methods based on analyses of temperature and precipitation have been often applied for productions of climate scenarios (e.g., ISIMIP3b https://www.isimip.org/gettingstarted/isimip3b-bias-adjustment/). However such methods do not ensure that other climate variables of selected models can well capture uncertainty ranges of larger ensembles (e.g., https://doi.org/10.2151/sola.2020-013 , https://doi.org/10.2151/sola.2021-009 , https://doi.org/10.2151/sola.2022-016). Therefore, at least, it is better to add a brief explanation of limitations of the method.*

We have added brief explanation of the limitations to the text.

*L339-342: Previous studies have suggested the linearity assumption does not hold for some cases (e.g., https://doi.org/10.1007/s10584-009-9765-1, https://doi.org/10.1175/2009JCLI3428.1, https://doi.org/10.2151/sola.2022-016). Did your method work well because you applied the pattern scaling only for ranges of 0.5K? (L 364-367)*

Yes, this was one motivation for carrying out the combination of time-shifting and pattern-scaling rather than just using pattern scaling – the linearity assumption holds better for small changes in temperature. We note that the methodology does not account for responses that depend on the history or rate of change of climate forcing, but this is a pragmatic approach where full transient climate simulations are not available.

*L369-373: Did you not produce climate scenarios of daily variables using the pattern scaling?*

Yes, we did produce scenarios of daily variables – we have clarified in the text.

**Response to RC2**

***Major comments***

*1. The biggest societal and economic impacts of climate change occur at the extremes. For example, heavy precipitation days that drive destructive flooding, or heat waves that drive human health impacts, agricultural desiccation, and high electricity use. The UK has seen both of these kinds of events in recent years. There is no evaluation of extremes in the present manuscript, only seasonal and longer term means. This is particularly problematic because some statistical downscaling*

*schemes have an easier time reproducing seasonal means than infrequent extremes. I request that the authors add some evaluations of extremes to the analysis. For example, 1-in-10 year values of daily precipitation in observations compared the downscaled and downscaled/bias corrected products over the historical period. Same for 1-in-10 year extreme daily maximum temperatures. (Or daily average if you don't have bias corrected Tmax.) I don't feel the data can be properly evaluated solely on the basis of the seasonal-mean results presented here.*

We agree that this is of value to potential users of the data - we have added an analysis of extreme values to the manuscript, including assessing 90th percentiles of daily mean temperature and 10-year return levels of precipitation against CHESS-met, as well as looking at the projected future change in 90th percentiles of daily maximum air temperature and 10-year return levels of precipitation.

**Minor comments**

*1. Many applications modelers assert they need hourly data for key variables rather than daily. Something to keep in mind for future efforts.*

Indeed. The choice of daily rather than hourly was based partly on the immediate needs of the project in which this was being used and partly on storage limitations (moving to sub-daily data would mean compromising further on the number of scenarios etc). When necessary, we use the data as an input to a land surface model, which runs at a half-hourly timestep and which does on-the-fly disaggregation from daily to half-hourly. This allowed us to balance the data storage requirements of increased temporal resolution against an increased number of ensemble members and scenarios.

*2. Line 49: "An overview of the benefits of higher resolution in simulated climate angle is presented in..." I don't know what the word "angle" is intended to mean here. Please replace with something else. Seems to me it should be more something like "results", or, even better, simply removed entirely.*

This is a stray word that has been removed.

*\* Line 49: "In many locations..." Do you mean in the UK specifically or generally around the globe? "...convective storms dominate" Dominate what? Please specify.*

We have replaced this sentence with "There are many regions globally where convective rainfall is more prevalent than large scale frontal rainfall, but to explicitly resolve the features of convective storms (rather than attempting parameterisation of their mean properties) needs a further step-change in resolution to kilometre scale."

*\* Line 62: "...but often storage limitations have precluded storing outputs at higher than a monthly timestep." This statement is dated and no longer correct. CMIP3 had primarily monthly output, but that was 18 years ago. Even CMIP5 from a decade ago had a large amount of daily data. The current version, CMIP6, has a large volume of daily data from numerous models and ensemble members. Personally I would have expected an effort such as the one described here to be pushing to sub-daily temporal resolution for selected 2-d surface variables.*

We have remove this paragraph and replaced it with a discussion of daily vs sub-daily data, and the motivation for our choice to produce daily.

*Line 83: "Pedde et al. have enriched and downscaled the global SSPs" ... I appreciate that you have supplied a reference for this assertion, but even so please expand briefly on "enriched" and "downscaled". Enriched in what way? How comparable are the results herein to those that would be obtained using standard SSP scenarios? This deserves a note in the text.*

The UK-SSPs are not used in this study, rather they are a companion data set that can be used alongside the CHESS-SCAPE climate data to explore possible future scenarios for the UK (eg Brown et al, 2022 https://doi.org/10.1029/2022EF002905 ). We have added to the text to clarify.

*Line 86: "..even with the same emissions scenarios, different climate models produce different projections of climate variables in the future, adding to the uncertainty." The role of model differences is rightly called out here, but it is an mistake to omit a mention of the key role of natural internal climate variability in producing different projections of future climate variables, particularly for a region as small (compared to the globe) as the UK. If you had 10 real Earths they would produce 10 different future climates for the UK just due to natural internal climate variability, and those differences can be significant for regions the size of the UK, even on decadal timescales. For an example over North America, see Deser et al. 2012 Nature Climate Change doi:10.1038/NCLIMATE1562.*

See next response.

*Line 93: "because the models are free-running, internal variability means that each realisation of the future climate will not necessarily replicate the historical period exactly -- the models may be 'biased' with respect to observations". I think this is a misstatement, I assume you meant to say "each realisation of the HISTORICAL climate will not necessarily replicate the historical period exactly". Otherwise I can't make sense of this statement. Also, it's a bit sloppy: biases and internal variability are not the same thing. Bias correction should not be used to "correct" a model that lacks biases but has differences from historical observations due only to a different trajectory of natural internal variability (e.g., Ayar and Mailhot 2021, Scientific Reports v 11 doi.org/10.1038/s41598-021-82715-1). But as a matter of practice it is often done since the two (biases vs. internal variability) can be hard to distinguish unless you have numerous ensemble members. The text should be more careful to point out these issues.*

Yes, this should refer to historical climate, not future. We have addressed both of these comments with updated and clarified text to address the issue of internal variability vs biases.

*Line 153: "However, they are not spatially coherent..." Please elaborate, in a sentence or two, in what sense they are not spatially coherent and why.*

We will add text to explain this further.

*Line 170: Was surface temperature corrected for the change in elevation, either before you obtained the data or by yourselves after you obtained the data? If so, how was it accomplished, using a constant lapse rate or a variable one? If a constant lapse rate, what value did you use? Please specify in the text. (Later note: I guess this is in Fig 2 and line 264 but it would be helpful to just mention it here.)*

In responding to this, we realised that the elevation was not used in the regridding of UKCP18. We have therefore noted this in the text and added a citation.

*Line 195: I spent some time trying to find the cited Spackman 1993 document and/or data on the web and was unsuccessful. Is this document actually available anywhere? The web page at https://catalogue.ceh.ac.uk/documents/2a2b1b05-a30f-4e04-9b37-75fa5ef5c26b seems to be an index page for this data but does not show where the data can be downloaded, nor does it have either a copy of Spackman 1993 or a link to that document. Anyway, is this actually just rainfall, as the title seems to imply, or does it include snowfall as well? Is snow simply ignored? The northern parts of Great Britain can average 30-60 days of snowfall per year. If snow is ignored, the text should state that.*

SAAR is a proprietary data set that was produced by the Met Office, along with the Spackman document. Licensing conditions mean that we are unable to share it (hence the index page not providing a download link). It is a predecessor of HadUK-Grid, which is freely available in the BADC data centre: https://catalogue.ceda.ac.uk/uuid/4dc8450d889a491ebb20e724debe2dfb . We chose to use SAAR in this work for consistency with other UKCEH products and models, which continue to make use of SAAR.

Snow (and other solid precipitation) is implicitly included, as SAAR is derived from the Met Office's station network, which defines rainfall to be "the amount of liquid precipitation plus the liquid equivalent of any solid precipitation (that is the liquid obtained by melting snow or ice that has fallen)" (MIDAS Data user guide, https://zenodo.org/record/7357325). The averages are calculated with and applied to the total without different treatments for different precipitation types. We have added this detail to the text to clarify.

*Line 202: I'll be honest that I keep getting surprised how old the reference data sets are. Rainfall from 1993? Wind from 1992 and 1985? I strongly suggest the authors use more up-to-date training data sets for any future work.*

The reason for using these particular data sets was for consistency with the historical CHESS-met data and other UKCEH products. Future work will indeed allow us update the methodology and inputs to create new historical and future climate data sets.

*Line 206: Are future changes in albedo due to projected reductions in snow cover ignored? If so, please specifically note this in the text. E.g., "Future changes in albedo due to projected reductions in snow cover are not included in this work."*

Yes, we are not taking into account changing albedo due to snow cover, so we have added this to the text.

*Line 234: "Five of the models..." Should change this to "Five of the ensemble members..." to be consistent.*

We have updated this in the text.

*Line 260: "All interpolation from 1 km to 12 km resolution..." It would be courteous to orient the reader better by adding a short phrase reminding them of the reason 1 to 12 km interpolation is performed, even though it's mentioned previously.*

Yes, we have updated the text here.

*Line 264: Please specify the constant physical lapse rate you used. Later note: I see that it's specified later, but it would be helpful to state it at this point.*

We have included the lapse rate value here.

*Line 276: I'm surprised you applied an offset to the wind speed instead of applying a factor. I think factors are more appropriate for positive definite variables like wind speed. Did you ever have the situation of producing a negative wind speed when using an offset? How did you handle that case? Please add to the text. Later note: doubly confusing because later on, around line 300, wind speed is scaled, not offset, in the bias correction process.*

Again, this was used for consistency with CHESS-met which used the offset method, chosen based on investigations when designing the product. We introduced a minimum threshold to ensure that we did not end up with any negative wind speeds. A scale factor was used for bias-correction and pattern-scaling to avoid any further possible negative wind speeds.

*Line 279: OK now I'm confused. At line 279 is says that the source of the bias correction data is CHESS-met. At line 195 it says that the precipitation was downscaled using Spackman 1993. Why didn't you use the CHESS-met data for downscaling as well as bias correction? Please elaborate in the text.*

The CHESS-met data were themselves interpolated (most variables from 40km MORECS data, the precipitation from station data) using a version of the same methodology and the same reference data sets. We applied this methodology to the new data set, rather than simply using CHESS-met itself for downscaling, to allow for differences between the MORECS observational data sets and climate model outputs. We have clarified this in the text.

*Line 286: "In general, the CHESS-SCAPE data is found to be robust in its ability to reproduce the average features of the CHESS-met data, and so our approach to bias correction is parsimonious". Statistical downscaling typically has an easy time reproducing the \*average\* features of a meteorological data set. However most of the impacts of climate are felt at the extremes, not in the average features. How well does CHESS-SCAPE do in reproducing, say, 20-year return values of daily*

*precipitation? Or once in a decade heat waves? Or rare cold snaps, which can have significant agricultural impacts? Extremes are both critical to impacts and a much more robust measure of the quality of statistically downscaled data than average features. I find this entire discussion too subjective ("very similar", "were different enough") and non-specific to be useful. Please add some figures showing the biases at the extremes.*

We have provided some plots and quantitative metrics to demonstrate the biases at the extremes, in particular 90th percentiles of daily mean and maximum air temperature, and 10-year return levels of precipitation.

*Line 303: Please cite in the text the source of the 400 W/m\*\*2 threshold, or otherwise explain why you picked it.*

This threshold was chosen to encompass the range of the historical CHESS-met data, we have added this to the text.

*Line 497: The lapse rate is specified in units of C/m\*\*2. Presumably the "m\*\*2" is a typo, should be C/m.*

Yes, this should be C/m – we have corrected this in the text.